# Plastic-Degrading Enzymes from Marine Microorganisms and Their Potential Value in Recycling Technologies

**DOI:** 10.3390/md22100441

**Published:** 2024-09-26

**Authors:** Robert Ruginescu, Cristina Purcarea

**Affiliations:** Department of Microbiology, Institute of Biology Bucharest of the Romanian Academy, 296 Splaiul Independentei, 060031 Bucharest, Romania; robert.ruginescu@ibiol.ro

**Keywords:** synthetic polymers, PET hydrolases, PLA biodegradation, cutinase, esterase

## Abstract

Since the 2005 discovery of the first enzyme capable of depolymerizing polyethylene terephthalate (PET), an aromatic polyester once thought to be enzymatically inert, extensive research has been undertaken to identify and engineer new biocatalysts for plastic degradation. This effort was directed toward developing efficient enzymatic recycling technologies that could overcome the limitations of mechanical and chemical methods. These enzymes are versatile molecules obtained from microorganisms living in various environments, including soil, compost, surface seawater, and extreme habitats such as hot springs, hydrothermal vents, deep-sea regions, and Antarctic seawater. Among various plastics, PET and polylactic acid (PLA) have been the primary focus of enzymatic depolymerization research, greatly enhancing our knowledge of enzymes that degrade these specific polymers. They often display unique catalytic properties that reflect their particular ecological niches. This review explores recent advancements in marine-derived enzymes that can depolymerize synthetic plastic polymers, emphasizing their structural and functional features that influence the efficiency of these catalysts in biorecycling processes. Current status and future perspectives of enzymatic plastic depolymerization are also discussed, with a focus on the underexplored marine enzymatic resources.

## 1. Introduction

Plastics are versatile, durable, and cost-effective materials that have become essential in various aspects of modern life. Since their widespread adoption in the 1950s, plastics have revolutionized nearly every sector of the economy, including packaging, construction, textile, automotive, consumer products, and healthcare [1]. Accordingly, annual plastic production continuously increased from 2 million metric tons (Mt) in 1950 to 400.3 Mt in 2022, surpassing most other materials [2,3]. Seven groups of fossil-based polymers accounted for three-quarters of global plastic production in 2022, namely polypropylene (PP) with 18.9%, low-density polyethylene (LD-PE) and linear low-density polyethylene (LLD-PE) with 14.1%, polyvinyl chloride (PVC) with 12.7%, high-density polyethylene (HD-PE) with 12.2%, polyethylene terephthalate (PET) with 6.2%, polyurethane (PUR) with 5.3%, and polystyrene (PS) with 5.2%. Additionally, other fossil-based plastics constituted 15.9% of the total share, while bio-based polymers accounted for 0.5% of the production, carbon-captured plastics represented less than 0.1%, and recycled plastics corresponded to 9% (8.9% mechanically recycled and less than 0.1% chemically recycled) [3].

The high chemical stability of the macromolecular structure that confers plastics versatility in a wide range of applications is also responsible for their resistance to abiotic and biological degradation. Consequently, it is estimated that about 91% of all plastic ever produced has accumulated in landfills, been incinerated, or been discarded in the natural environment, while around 9% has been recycled [2,4]. Of the four disposal options, incineration generally produces the largest amounts of greenhouse gases, thus having a large impact on climate change [5]. Plastics that end up in landfills or are released into the environment in an uncontrolled manner can take hundreds or even thousands of years to degrade [6]. During their slow decomposition, they release microplastics (i.e., particles smaller than 5 mm) and other toxic compounds, such as heavy metals (e.g., cadmium used in pigments) [7] and additives (e.g., phthalates), [8] that can affect soil and water quality, marine and terrestrial wildlife, and potentially human health through the food chain [5].

The long-term environmental pollution caused by plastic waste prompted authorities to develop international strategies aimed at enhancing recycling processes and reducing fossil-based plastic production in favor of biodegradable bio-based feedstocks. The European Union, for instance, supports the principles of a circular economy, where plastics are reused and recycled rather than discarded into landfills or incinerated, thus promoting a sustainable model by reducing waste, resource use, and greenhouse gas emissions [9,10].

Recycling of fossil-based plastics is primarily done by mechanical and chemical processes (Figure 1). Mechanical recycling is the most common method and involves sorting, grinding, washing, melting, and pelletizing plastic waste. This process allows plastic waste to be recycled multiple times, albeit with a progressive loss of quality caused by the thermal-mechanical degradation of polymers during reprocessing [11,12]. Mechanical recycling is generally used for PET and PE due to their chemical, mechanical, and thermal properties, which make them suitable for the process. In contrast, temperature-sensitive plastics and those resistant to flow at high temperatures, such as thermosets, cannot be recycled this way [13].

Chemical recycling complements mechanical methods by enabling the recycling of complex, mixed, and contaminated plastics. This method includes processes like pyrolysis, gasification, solvent-based purification (dissolution), and depolymerization, which convert plastic waste into simpler molecules (i.e., monomers, oligomers, polymers) that can be used for various practical purposes, including creating value-added products (i.e., upcycling) and new plastics with properties equivalent to virgin materials [14,15,16,17,18,19]. Although chemical methods are superior to mechanical recycling, they require harsh physicochemical conditions, such as high temperature (>180 °C), high pressure (20–40 atm), and chemical catalysts (e.g., caustic bases, concentrated sulfuric or nitric acids) that result in large energy requirements and hazardous waste generation. In an attempt to overcome these limitations, researchers have recently focused their attention on developing recycling strategies based on enzymatic, photo-, electro-, and microwave-assisted catalysis under environmental and relatively mild conditions [15,20,21], as well as on finding and genetically engineering plastic-degrading microorganisms [15,18]. Although these emerging technologies may offer a more environmentally friendly and energy-efficient alternative to conventional chemical recycling processes, many are still in the laboratory stage and require further research to achieve industrial applications [20].

A third method for recycling plastics is biological or organic degradation (Figure 1), which utilizes microorganisms and their enzymes to decompose biodegradable polymers into environmentally friendly compounds, such as gases (e.g., carbon dioxide, methane) and other organic molecules (e.g., monomers, oligomers) that may be used for recycling and upcycling purposes [9,22]. Biological degradation is a complex process typically involving from two to four stages: (*i*) the attachment of microorganisms to the polymer surface (biofilm formation), (*ii*) the synthesis of extracellular enzymes that depolymerize polymeric chains into oligomers and monomers (depolymerization), (*iii*) the transportation of these low molecular weight molecules into the cells (assimilation), and (*iv*) the metabolization of these molecules to produce energy, water, and other simple compounds, such as carbon dioxide and methane [23,24]. This method is sustainable and energy efficient, but it applies to a limited number of polymer types (both bio- and fossil-based), mainly polylactic acid (PLA), polyhydroxyalkanoate (PHA), polybutylene succinate (PBS), polycaprolactone (PCL), polybutylene adipate terephthalate (PBAT), and starch, which constituted approximately 0.29% of the total amount of plastics produced in 2022 [22].

While the first reports of microbial strains capable of degrading aliphatic synthetic polyesters, such as PCL, date back to the 1970s [25,26], the role of enzymes in plastic recycling began to be recognized in 2005 when Müller and colleagues [27] reported the enzymatic hydrolysis of PET, an aromatic polyester regarded as enzymatically inert at that time. Since then, research on enzymatic catalysis of plastic has increased sharply [28], and the resulting high influx of data has necessitated the creation of databases (e.g., PlasticDB [29] and PAZy [30]) to compile and utilize the extensive information effectively. For instance, as of June 2024, the database PlasticDB [31] lists 753 species of microorganisms and 219 enzymes that break down plastics. PET and PLA have been the focus of most enzymatic depolymerization studies, and, thus, the largest fraction of plastic-degrading enzymes characterized to date are specific to these two polymers. Moreover, based on the knowledge accumulated over the past two decades, the French company Carbios has brought to maturity a technology for enzymatic depolymerization and recycling of PET, culminating in the announcement that an industrial plant utilizing this technology will be constructed by 2025 [4,32]. Nevertheless, the enzymatic degradation of recalcitrant polymers, such as PE, PP, PVC, PS, and PUR, has achieved only limited success, and much more research is necessary to identify efficient biocatalysts against plastics with high chemical stability [4].

Plastic-degrading enzymes, also known as plastic hydrolases or depolymerases, typically break down synthetic polymers into their monomers or oligomers through hydrolysis. These enzymes are categorized into several types based on their substrate specificity and reaction mechanism. The main types include cutinases (EC 3.1.1.74), which degrade cutin (the polyester structural component of the plant cuticle) as well as synthetic polyesters; carboxylesterases (EC 3.1.1.1), which act on ester bonds in various polyesters; triacylglycerol lipases (EC 3.1.1.3), which break down triglycerides but also exhibit activity against certain plastics; and PET hydrolases (EC 3.1.1.101), specifically targeting PET polymers [4]. Most of these enzymes originate from prokaryotic microorganisms, particularly thermophilic sources, due to the need for most plastics to be efficiently degraded at high temperatures (>65 °C), close to their glass transition temperatures [33]. Some notable examples of thermostable cutinases active on PET have been derived from actinomycetes species, such as *Thermobifida* (e.g., TfH/BTA-1 [27]), *Thermomonospora* (e.g., Tcur0390 [34]), and *Saccharomonospora* (e.g., Cut190 [35]), as well as from plant compost metagenomes (e.g., LC-cutinase/LCC [36] and PHL7 [37]). Several heat-labile plastic-degrading enzymes derived from mesophilic and cold-adapted microorganisms, such as *Is*PETase isolated from *Ideonella sakaiensis* [38] and OaCut from *Oleispira antarctica* [39], have also been reported in the literature, serving as a base for protein engineering studies aimed at enhancing their thermal stability [40].

The environments from which plastic-degrading enzymes have been obtained are diverse, including soil [38,41], compost [36], surface seawater [42], and habitats with extreme physicochemical conditions, such as hot springs [43], hydrothermal vents [44,45], deep-sea [45], and Antarctic seawater [39]. This high diversity of environments in which plastic-degrading microorganisms can survive is correlated with the adaptation of their enzymes to function in a wide range of physicochemical conditions, including extreme temperature, pH, salinity, and pressure. In this regard, enzymes derived from marine microorganisms, particularly those inhabiting extreme habitats, may present extreme features, such as high thermostability, activity under acidic pH, and resistance to salts and pressure, which can be advantageous in industrial plastic-recycling processes [4,21].

Several excellent reviews have been published that comprehensively discuss the recent advances in enzyme-based depolymerization of plastics [4,21,46,47], with some focusing on marine [48] and extremophilic microorganisms [49]. To the best of our knowledge, no recent study has reviewed plastic-degrading enzymes from marine environments, albeit Lv and colleagues [48] concentrated more on marine microorganisms and metabolic pathways involved in plastic degradation rather than on the structural and functional characteristics of their enzymes. Considering the recently registered progress in this area, the present work aims to review the latest advancements in marine-derived enzymes capable of depolymerizing synthetic plastic polymers. This review highlights the unique characteristics of these enzymes that potentially make them advantageous in the harsh conditions of industrial plastic-recycling processes, as well as their current limitations.

## 2. Polyethylene Terephthalate (PET) Depolymerization

### 2.1. About PET and Its Recycling Strategies

Polyethylene terephthalate (PET) is a semiaromatic and semicrystalline polyester consisting of repeating units of terephthalic acid (TPA) and ethylene glycol (EG). Characterized by high mechanical strength, gas and liquid impermeability, high optical clarity, and low weight, PET is primarily used to produce textile fibers, single-use bottles, and rigid containers. Generally, TPA and EG are made from petroleum resources, although bio-based PET synthesis is possible by combining biomass-derived EG and fossil-based TPA [4].

Due to its wide utilization in short-life products, PET is the most commonly recycled plastic globally, primarily through mechanical processing [4]. However, major drawbacks are associated with mechanical recycling, namely, (*i*) only transparent and pure PET waste can be used to produce bottles; (*ii*) PET ductility decreases significantly after the first cycle of recycling, mainly due to the thermo-mechanical and thermo-oxidative degradation of polymers, necessitating the use of recycled PET for lower-value products, such as fibers in carpeting, which cannot be recycled further; and (*iii*) contaminants, such as traces of PLA and PVC, adversely impact the quality of recycled PET [50].

A less exploited recycling strategy involves the chemical depolymerization of PET chains into their highly thermostable original monomers (i.e., TPA and EG) or intermediate oligomers, such as bis(2-hydroxyethyl) terephthalate (BHET) and mono(2-hydroxyethyl) terephthalate (MHET) (Figure 2), which can be reprocessed into new products with characteristics similar to virgin materials. The various depolymerization processes developed to date utilize different reagents and reaction conditions (e.g., alcoholysis, nitrogen-based solvolysis, glycolysis, and alkaline, acidic, neutral, or enzymatic hydrolysis), each with its advantages and drawbacks [51,52,53]. Enzyme-based PET depolymerization emerged as a promising alternative among the various chemical recycling methods. This is primarily due to (*i*) the milder temperatures required for enzymatic depolymerization, which results in less energy consumption; (*ii*) the unnecessity of using hazardous chemicals such as highly polar solvents and concentrated acids; and (*iii*) the high selectivity of enzymes for polymer ester bonds facilitates monomers recovery and allows the process to apply to blended materials, avoiding intensive plastic sorting [4,54,55].

However, PET enzymatic depolymerization is not an easy task, facing major challenges primarily due to the polyester’s macromolecular architecture (e.g., crystallinity and surface hydrophobicity), which restricts enzyme binding, and, secondarily, due to the limitations of the biocatalysts (e.g., thermolability, low catalytic efficiency, and enzyme inhibition). Polymer crystallinity is among the key factors affecting PET depolymerization [56]. As a semicrystalline polymer, PET is characterized by regularly arranged, tightly packed regions (crystalline) and nonorganized, less cohesive regions (amorphous). The amorphous regions comprise fractions that are mobile at temperatures above the PET’s glass transition temperature (i.e., 70–80 °C) and fractions that maintain rigidity, connecting crystalline regions with mobile amorphous domains (Figure 2) [4,57]. Although some studies have claimed that semicrystalline PET polymers (i.e., 30–40% crystallinity) are susceptible to enzymatic depolymerization [38,58], several recent reports have unequivocally demonstrated that only amorphous PET, particularly the mobile regions, can be subjected to enzyme attack [32,40,59]. Consequently, until novel enzymes capable of acting on semicrystalline polymers are discovered or engineered, the current technology relies on PET pretreatments through various physical (e.g., extrusion [32]) or chemical (e.g., solubilization by organic solvents [60]) methods to increase its amorphous fraction (amorphization). Subsequently, this step is followed by enzymatic depolymerization at temperatures near the PET’s glass transition temperature, at which chains are mobile, favoring enzyme binding. It is thus mandatory that PET hydrolases exhibit high thermostability above 65 °C [4,32,56].

Extensive efforts have been made in recent years to discover and characterize PET-active enzymes, as well as to engineer their structure to increase thermostability, improve their interaction with the substrate, optimize catalytic activity, and minimize their inhibition by degradation intermediates [4,21]. However, there remains room for improvement, and bioprospecting studies to discover more efficient biocatalysts from various environments are still of particular interest, as genes encoding these enzymes are spread across numerous bacterial and fungal branches [61]. The next section will focus on PET hydrolases derived from marine environments and their potential utility in PET recycling.

### 2.2. PET-Depolymerizing Marine Enzymes

Numerous PET-depolymerizing enzymes (hereafter also referred to as PET hydrolases, without specific reference to the EC class 3.1.1.101) from various bacterial and fungal sources have been discovered and characterized, as detailed in several recent reviews [4,21,46,47]. Of the 110 PET-degrading hydrolases listed in the PAZy database [62] as of June 2024, 15 (excepting engineered variants) originate from marine sources, either cultured microorganisms or environmental metagenomic datasets (Table 1). As discussed further in this section, most of these enzymes belong to a subclass of the α/β-hydrolase enzyme superfamily, characterized by a highly conserved core domain comprising an eight- or nine-stranded β-sheet surrounded by six to eight α-helices. Notably, two enzymes, Rcut and PET46, exhibit significantly different structural properties. Nevertheless, despite their differences in amino acid sequence and three-dimensional conformation, all PET hydrolases share a common catalytic machinery with a highly conserved catalytic triad (Ser, His, Asp), which performs the ester bond hydrolysis through a nucleophilic attack by the catalytic serine’s oxygen on the carbonyl carbon atom present in the scissile ester bond [4].

PET hydrolases have been classified into two main types, type I and type II, with type II further divided into subtypes IIa and IIb. This classification is based on comparisons to the primary structure of *Is*PETase, a type IIb enzyme derived from *Ideonella sakaiensis* [74]. Among the 15 PET-degrading enzymes of marine origin, eight are classified as type IIa and two as type I. The remaining five enzymes are unclassified due to significant divergence in their amino acid sequences compared to the type I and type II enzymes (Figure 3). Taxonomically, these enzymes are distributed across various phyla: ten are associated with Pseudomonadota, three with Actinomycetes, one with Bacteroidota, and one with the archaeal phylum Bathyarchaeota (Table 1 and Figure 3).

Since most of the marine-derived PET hydrolases originate from cold habitats, such as Antarctic seawater and deep-sea sediments, they were found to be adapted to function optimally at relatively low temperatures (i.e., 25–50 °C), featuring melting temperatures (T_m_) between 40 and 50 °C. The only notable exception is PET46, which was derived from sediments near a hydrothermal vent and exhibited activity and stability at high temperatures. Generally, these enzymes performed optimally at a slightly alkaline pH (7.4–9.0), and some (i.e., PET6 and SM14est) were remarkably tolerant to high salt concentrations (Table 1).

Accurately comparing the enzymes’ efficacy in PET depolymerization, based on the total amount of hydrolysis products released from PET, was hindered by the use of substantially different substrates. Nonetheless, three wild-type enzymes (PET46, Ple629, and PET6) and three chimeric variants (CM^A266C^, PET6-VSTA, and PET6-ExLoop) produced over 1 mM of hydrolysis products (Table 1), highlighting their potential for large-scale recycling applications. A detailed review of each marine-derived enzyme listed in Table 1 is provided below.

An extensive screening of protein databases and metagenomic datasets using a Hidden Markov Model (HMM) approach has led to the identification of 853 putative PET hydrolases, including the marine microorganism-derived enzymes PET5 and PET6 [41]. Since these enzymes produced clearing halos on agar plates containing PET nanoparticles or PCL after their cloning and heterologous expression in *Escherichia coli* [41], they were chosen for further biochemical characterization [39,64].

PET5, also known as OaCut [39], was derived from the genome of *Oleispira antarctica* RB-8, a psychrophilic hydrocarbonoclastic bacterium isolated from the superficial seawater of Rod Bay, Antarctica [76]. Computational analysis of the sequence and structure of OaCut revealed common features with other type IIa PET hydrolases [74], namely, two disulfide bridges between Cys220-Cys257 and Cys291-Cys308, a Phe256 residue at the corresponding position of Ser238 in *Is*PETase, and an extended loop with three extra amino acids (Gly-Ser-Ile) between position 262 and 264 [39]. The presence of two disulfide bridges is generally associated with enhanced thermal stability in enzymes [74]. However, OaCut exhibited lower stability at elevated temperatures compared to type I PET hydrolases, which possess only one disulfide bond. This observation suggested that the high thermal stability of type I PET hydrolases is attributed to structural features other than disulfide bonding [74]. Moreover, the melting temperature (T_m_) of OaCut was 40.4 °C, reflecting its moderate thermal stability typical of enzymes derived from psychrophilic microorganisms. By assaying the ability of OaCut to hydrolyze amorphous PET films at 25 °C, a modest weight loss of 0.4% was observed after 6 days. This efficiency is five times lower than that of Mors1, which was tested under the same conditions (Table 1) [39].

PET6 was identified by screening the genome of *Vibrio gazogenes* DSM-21264, a marine bacterium isolated from sulfide-containing mud collected from a saltwater marsh [41,77]. The crystal structure of PET6, resolved at 1.4 Å resolution, revealed an α/β-hydrolase fold characterized by eight β-strands surrounded by α-helices and a conserved catalytic triad (Ser163, Asp209, and His241) (Figure 4). It shared common features with other type IIa PET hydrolases, as detailed above for PET5, but with the major difference that PET6 exhibited the formation of a third disulfide bridge near the N-terminus, a rare trait among PET hydrolases [64]. This additional disulfide bond was also observed in Mors1 [63] and was suggested to be an adaptation to low temperatures [39]. Moreover, the ability of PET6 to bind three monovalent ions (two sodium ions and a chloride ion) instead of divalent ions, as in the case of many PET hydrolases, was considered an adaptation to saline environments with high sodium chloride concentrations [64].

The purified heterologous PET6 enzyme was shown to optimally hydrolyze post-consumer PET (around 10% crystallinity) at 45–50 °C in the presence of 1–1.5 M NaCl. Increasing salt concentrations from 50 mM to 1.5 M promoted catalytic activity by stabilizing the proteins, as the T_m_ increased from 49.8 °C at 50 mM NaCl to 57.7 °C in the presence of 1 M NaCl. Under optimal conditions, the hydrolysis products, including TPA, BHET, and MHET, had a summed concentration of approximately 1.1 mM (Table 1). This is comparable to the concentration produced by the chimeric Mors1 variant (CM^A266C^) under a similar reaction temperature and pH [63]. However, this comparison may not be entirely appropriate, as the PET substrates used in the two studies differed.

Three PET6 variants, namely, PET6-YLA, PET6-ExLoop, and PET6-VSTA, were created by introducing mutations into the wild-type protein sequence inspired by the structure of *Is*PETase [64]. Specifically, in the PET6-YLA variant, the tyrosine residue at position 248 was replaced by an alanine (mutation Y248A). The PET6-ExLoop variant involved replacing the entire extended loop composed of TGYPSE residues between positions 246–251 with the SGNSNQ sequence from *Is*PETase. In the PET6-VSTA variant, two mutations were introduced: valine at position 91 was replaced with threonine (mutation V91T), and serine at position 92 was replaced with alanine (mutation S92A). Of the three variants, PET6-ExLoop and PET6-VSTA outperformed the wild-type PET6 by 79% and 58%, respectively, at 50 °C and 1 M NaCl (Table 1). Under these reaction conditions, PET6-VSTA demonstrated a 1.6-fold higher release of hydrolysis products from PET substrate compared to *Is*PETase. However, this performance was approximately three times lower than that of *Is*PETase at 30 °C and 1 M NaCl [64].

Mors1 was identified in the genome of *Moraxella* sp. TA144, a psychrophilic bacterium isolated from Antarctic seawater [78], through a BLAST search of the UniProtKB database using the amino acid sequence of *Is*PETase as a query [39]. This search, which revealed a 45% sequence identity with *Is*PETase, was followed by the cloning and expression of the enzyme-coding gene in *E*. *coli*. The purified enzyme completely clarified a PCL nanoparticle suspension at an optimal temperature of 25 °C, with its preference for moderate temperatures also highlighted by its relatively low T_m_ of 52 °C. Moreover, the enzymatic activity was optimal at an alkaline pH of 8.0 and was enhanced by 20% by adding 200 mM NaCl to the reaction buffer. The promoting effect of salts on enzymatic activity has also been reported in other PET hydrolases, such as *Is*PETase [79], PET6 [64], and SM14est [71], and can be attributed to the formation of coordinated bonds between certain enzyme amino acids and Cl^−^ anion, which stabilizes the catalytic center of the enzyme [79]. By assaying the ability of Mors1 to hydrolyze amorphous PET films at 25 °C, a weight loss of 1.98% and 2.5% was observed after 6 days and 10 days, respectively. The hydrolysis products released after 24 h, represented by MHET and TPA, had a summed concentration of about 0.26 mM, similar to that produced by *Is*PETase under the same conditions; however, the concentration of Mors1 used was four times higher than that of *Is*PETase (Table 1) [39]. The crystal structure of Mors1 (Figure 4), which was solved at 1.6 Å resolution, revealed high similarities with other PET hydrolases [63]. Specifically, it showed a canonical α/β-hydrolase fold comprised of a central β-sheet of nine strands surrounded by six α-helices, a conserved Ser-His-Asp catalytic triad, and a binding subsite II almost identical to that of other type IIa PET hydrolases. More notable differences were observed in subsite I, where two conserved residues were replaced in Mors1 by Tyr214 and Asp153, and in the N-terminal region, where a third disulfide bond was confirmed between residues Cys60 and Cys109 [63]. This third disulfide bridge was also reported in PET6 [64] and could represent a stabilization strategy of cold-adapted microorganisms to counterbalance the flexibility of their enzymes [39].

Two engineered variants of Mors1, termed CM and CM^A266C^, have been obtained by substituting the 15-residue region that comprises the highly flexible active site loop in Mors1 with a homologous 13-residue region from the highly efficient and thermostable LCC [63]. The chimeric variant CM lost the active site disulfide bridge (Cys231-Cys266) due to the absence of cysteine residues in the homologous loop of LCC, resulting in a complete loss of enzymatic activity. In contrast, the CM^A266C^ chimera, which had its active site disulfide bridge restored by replacing alanine with cysteine at position 266, exhibited a shift in optimal activity temperature from 25 °C to 45 °C and a fivefold increase in catalytic activity against amorphous PET films at 45 °C compared with wild-type Mors1 activity at 25 °C (Table 1) [63]. Although CM^A266C^ exhibited significant improvements over Mors1, its activity is still 30–40 times lower than that of thermophilic PET hydrolases LCC and PHL7, which were reported to achieve 73% and 100% weight loss of amorphous PET films within 24 h of reaction at 70 °C [37].

The Ple628 and Ple629 encoding genes were recovered from the metagenome of a marine microbial consortium capable of utilizing a commercial PBAT-blend film as the sole carbon source [66]. These two tandem genes clustered phylogenetically with cutinase-encoding genes from *Marinobacter* species and shared over 50% sequence similarity with *Is*PETase [66]. Considering their putative plastic-degrading capabilities, they were subsequently cloned and expressed in *E*. *coli*, purified, and structurally and functionally characterized [65]. The sequences and crystal structures of Ple628 and Ple629 enzymes revealed high similarity to other type IIa PET hydrolases. They were characterized by a central twisted β-sheet composed of nine β-strands surrounded by eight α-helices. The catalytic triad Ser-Asp-His, conserved among all PET hydrolases, was also identified in both enzymes at positions 174/179-220/225-252/257 (positions in Ple628/Ple629, respectively). Moreover, two disulfide bridges were observed in each enzyme: Cys217-Cys254 and Cys288-Cys305 in Ple628 and Cys222-Cys259 and Cys297-Cys314 in Ple629. The presence of two disulfide bonds, a common trait of type IIa PET hydrolases of mesophilic and psychrophilic origin, and the relatively low T_m_ (Table 1), suggested that Ple628 and Ple629 are adapted to moderate temperatures. Indeed, the optimal activity of both enzymes was determined at 30 °C, and they retained over 58% of their maximal activity at 20 °C. However, at 40 °C, the activity of Ple628 and Ple629 dropped to 31% and 15% of that detected at 30 °C, respectively. When assaying the ability of these enzymes to hydrolyze PET nanoparticles of unspecified crystallinity at 30 °C, Ple629 released a 24-fold higher amount of MHET and TPA compared to Ple628 (Table 1) [65]. This level of product release is comparable to the concentrations produced by CM^A266C^, PET6-VSTA, and PET46, albeit under higher reaction temperatures (i.e., 45 °C, 50 °C, and 60 °C, respectively) and different PET substrates (Table 1).

PpelaLip was identified through a homology-guided sequence search of various extracellular hydrolases from *Pseudomonas* sp., using the amino acid sequence of *Thermobifida cellulosilytica* cutinase (Thc_Cut1) as a template [67]. The gene encoding PpelaLip was discovered in the genome of *Halopseudomonas pelagia* DSM-25163 (formerly *Pseudomonas*) [67], a psychrotrophic strain isolated from a culture of the Antarctic green alga *Pyramimonas gelidicola* [80]. Despite sharing only 12% sequence similarity with Thc_Cut1 (an enzyme known to efficiently hydrolyze various polyesters), the recombinantly produced and purified PpelaLip was able to degrade an amorphous, laboratory-synthesized PET-type polyester at 28 °C. The amount of released hydrolysis products (~17 µM TPA) was comparable to that produced by PE-H (20 µM MHET) but substantially lower than that of other marine-derived PET hydrolases, such as Ple629 (1500 µM TPA and MHET) and PET27 (872 µM TPA), that operated under similar reaction temperatures and pH, albeit on different PET substrates (Table 1). While PpelaLip has also been shown to hydrolyze polyoxyethylene terephthalate (PET-PEO) under realistic wastewater treatment plant conditions [81], a more comprehensive structural and functional characterization of this enzyme is currently lacking.

The gene encoding PE-H was identified in the genome of *Halopseudomonas aestusnigri* VGXO14^T^ (formerly *Pseudomonas*) [82], a marine hydrocarbonoclastic bacterium isolated from crude oil-contaminated intertidal sand [83]. This strain demonstrated the ability to hydrolyze various polyesters [84], prompting further investigation of its enzymatic capabilities. Consequently, the PE-H gene was cloned and expressed in *E. coli*, and the recombinant protein was purified and characterized both structurally and functionally. Notably, PE-H became the first type IIa PET hydrolase for which the crystal structure was solved [68]. The enzyme structure at 1.09 Å resolution showed a canonical α/β-fold with a central twisted β-sheet composed of nine β-strands surrounded by seven α-helices (Figure 5A). As with all type IIa PET hydrolases, PE-H showed a conserved catalytic triad (Ser171, Asp217, His249), two disulfide bonds (Cys214-Cys251 and Cys285-Cys302) for improved stability of the active site, and an extended loop region (Gly254, Gly255, Ser256) to facilitate interaction with the substrate. These structural features, along with the T_m_ of 51 °C, suggested that PE-H is adapted to moderate temperatures. Indeed, this enzyme has been shown to partially degrade an amorphous PET film at 30 °C (Table 1) but lacked activity on a more crystalline PET film from a commercial PET bottle. The sole hydrolysis product released was MHET, at a concentration of about 20 µM [68]. In comparison, other marine PET hydrolases that operated under similar reaction conditions, such as Ple629 and PET27, produced 1500 µM and 872 µM hydrolysis products, respectively (Table 1).

In an attempt to improve the activity of PE-H, a series of single and multiple amino acid substitutions were introduced into its sequence through site-directed mutagenesis inspired by the structure of *Is*PETase [68]. Out of the 12 mutants generated, only one variant (PE-H^Y250S^) exhibited enhanced activity. This variant was obtained by replacing the aromatic tyrosine residue at position 250, located adjacent to the histidine residue of the catalytic triad, with a smaller serine residue (mutation Y250S). The structural analysis of this variant revealed a rearrangement of the loop connecting β3-α2, resulting in an active site cleft much deeper compared with that of the wild-type PE-H (Figure 5B). This additional space created a more accessible active site, increasing the amount of MHET released from an amorphous PET film by 1.3-fold compared to the wild-type enzyme. Moreover, PE-H^Y250S^ displayed modest activity on semicrystalline PET film derived from a commercial PET bottle (Table 1) [68].

Another study by Erickson and collaborators [43] combined an HMM approach with machine learning to mine protein databases and metagenomic datasets for novel PET hydrolases and to predict the optimal temperature of the identified enzymes based on their sequence. From this analysis, the researchers selected 74 putative thermotolerant PET hydrolases for heterologous expression in *E*. *coli*, followed by purification and experimental screening. Of these, four enzymes, designated 403, 409, 412, and 606, were derived from marine sources. The genes encoding enzymes 403 and 409 were identified in a metagenomic dataset derived from seawater collected at a depth of 700 m in the Pacific Ocean and were putatively assigned to *Ketobacter* species. The gene encoding enzyme 412 was discovered in the genome of *Ketobacter alkanivorans* GI5^T^, a bacterial strain isolated from surface seawater in Garorim Bay, Republic of Korea [85]. The gene encoding enzyme 606 was identified in the genome of *Marinactinospora thermotolerans* DSM-45154, a moderately thermotolerant actinomycete isolated from deep-sea sediment collected at a depth of 3865 m in the northern South China Sea [86]. The purified recombinant enzymes 403, 409, 412, and 606 demonstrated partial hydrolysis of amorphous PET film at 60–70 °C, releasing the aromatic products BHET, MHET, and TPA at total concentrations of 9, 50, 11, and 345 µM, respectively (Table 1). However, these product yields were substantially lower than that of the thermophilic cutinase LCC, which produced over 27 mM of hydrolysis products under similar reaction conditions [43]. Of the four proteins, the crystal structure was only determined for enzyme 606. It exhibited an α/β-fold, featuring a highly conserved core domain with a 9-stranded β-sheet flanked by 8 α-helices. The structure revealed a conserved catalytic triad (Ser130-Asp176-His208) and a unique C-terminal disulfide bridge (Cys241-Cys259), while lacking the extended loop characteristic of type II PET hydrolases [43]. Based on these structural features, enzyme 606 can be classified as a type I PET hydrolase, a group that typically includes highly thermostable hydrolases with T_m_ above 70 °C, such as LCC and PHL7 [37]. Despite this classification, enzyme 606 exhibited a T_m_ of 53.9 °C [43], similar to other type II marine-derived enzymes, such as Mors1, PET6, and PE-H (Table 1), suggesting that it performs optimally at moderate temperatures rather than at temperatures near to the polymer glass transition temperature.

Eight months before the publication of Erickson et al.’s study [43], Liu and colleagues [69] reported the identification and characterization of *Mt*Cut, a PET hydrolase encoded in the genome of *Marinactinospora thermotolerans* DSM-45154. This enzyme (Genbank accession no: SJZ42839.1) has an identical predicted amino acid sequence to the enzyme later designated as 606 (Genbank accession no: WP_078759821.1) in Erickson et al.’s study [43], although they did not explicitly reference the earlier work on *Mt*Cut. The purified heterologous *Mt*Cut exhibited optimal hydrolytic activity against PET microparticles (42% crystallinity) at 45 °C, pH 8–8.5, and in the presence of Ca^2+^ at a 10–100 mM CaCl_2_ concentration. Under these conditions, the total amount of released TPA, MHET, and BHET was approximately 400 µM (Table 1). Notably, this enzyme performed better on 42%-crystallinity PET microparticles than on 10%-crystallinity PET films, as also reported for enzyme 606 [43]. This may be due to the enzyme’s active site cleft, enabling the polymer to adopt low-energy conformations in regions where monomers show a more linear arrangement, as in the case of high-crystallinity PET [43]. Moreover, *Mt*Cut activity and thermostability were enhanced by Ca^2+^, as evidenced by the T_m_ shift from 33 °C without calcium to 41.5 °C at 300 mM CaCl_2_. This feature was also reported for Cut190, a thermophilic microorganism-derived PET hydrolase whose tertiary structure slightly changed in the presence of calcium ions [87]. Another notable characteristic of *Mt*Cut is that it was not inhibited by released MHET, unlike other well-characterized PET hydrolases, such as *Is*PETase and ICCG (an LCC variant with improved activity and thermostability). This lack of inhibition may be due to *Mt*Cut’s ability to rapidly degrade MHET [69].

The Rcut encoding gene was identified in the genome of *Rhodococcus* sp. RosL12, a marine bacterial strain isolated from the Ross Sea, Antarctica [70]. Computational analysis of the amino acid sequence and predicted three-dimensional structure of Rcut revealed higher similarity to fungal enzymes than to bacterial ones (Figure 3). The structure exhibited an α/β-fold with a five-stranded β-sheet surrounded by four α-helices. It also featured a conserved catalytic triad (Ser114-Asp181-His194) and two disulfide bonds positioned at both termini of the protein (Cys32-Cys103 and Cys177-Cys184) [70]. Consequently, when compared to the other marine-derived PET hydrolases detailed in the present review, Rcut belongs to a different subclass of the α/β-hydrolase fold enzyme superfamily, the same subclass in which fungal PET hydrolases are classified [4]. The purified heterologous enzyme decomposed a PCL film at an optimal temperature of 40 °C and pH 9.0. However, only traces of TPA and MHET were released from PET film [70], suggesting that Rcut functions more as a PET surface-modifying enzyme rather than a true PET hydrolase [4].

The SM14est encoding gene was identified in the genome of *Streptomyces* sp. SM14, a bacterial strain isolated from the marine sponge *Haliclona simulans* [88]. Computational analysis of the amino acid sequence and predicted three-dimensional structure of SM14est revealed a higher level of similarity to thermophilic microorganisms-derived type I PET hydrolases than to marine-derived type II enzymes [71]. Specifically, SM14est showed 46% sequence identity and 79.4% similarity with PHL7, a type I PET hydrolase derived from plant compost [37]. In contrast, it showed 34.4%/75.8%, 33.6%/73.6%, and 32.2%/72.8% sequence identity/similarity with the marine-derived type II enzymes PET6, PE-H, and Ple629, respectively [71]. The predicted three-dimensional structure of SM14est exhibited an α/β-fold with a nine-stranded β-sheet surrounded by seven α-helices. The catalytic triad found in all PET hydrolases was also conserved in SM14est (Ser156, Asp202, His234). Consistent with other type I PET hydrolases, SM14est lacked the three extra amino acids of the loop connecting β8 and α6, which is thought to facilitate enzyme interaction with the polymer chain. However, the major structural difference between SM14est and other PET hydrolases was that SM14est lacked any disulfide bridges, whereas other PET hydrolases have been shown to possess from one to three such structures [88]. Despite lacking any disulfide bonds, SM14est exhibited the highest T_m_ (55 °C) [71] among the wild-type PET-depolymerizing enzymes derived from marine bacteria (Table 1). However, its T_m_ was still considerably lower than that of thermostable type I PET hydrolases, such as LCC and PHL7, whose T_m_ values exceed 79 °C [37].

The SM14est encoding gene was cloned and expressed in *Bacillus subtilis*, and the purified enzyme was assayed on semicrystalline PET powder with over 40% crystallinity [71]. Consistent with its T_m_, SM14est exhibited optimal activity at 45 °C, producing TPA, MHET, and BHET at a total concentration of 270 µM (Table 1). This amount of released products was lower than that reported for *Mt*Cut (400 µM) under similar reaction conditions. However, *Mt*Cut required a longer incubation time (Table 1), which suggests a potentially higher turnover rate for SM14est. Moreover, the enzymatic activity of SM14est was significantly influenced by salts, showing a 5-fold higher amount of released hydrolysis products in the presence of 0.5 M NaCl compared to in its absence [71]. This behavior is consistent with that reported for PET6 [64], which also showed an increase in its activity and thermal stability in the presence of up to 1.5 M NaCl. Similar to PET6, the salt tolerance of SM14est has been suggested to be attributed to the formation of coordinated bonds between certain enzyme amino acids and three monovalent ions (two sodium ions and a chloride ion), which together stabilize the catalytic center of the enzyme [64,71].

The PET27 encoding gene was identified in the genome of *Aequorivita* sp. CIP 111184 [72], a bacterial strain isolated from Antarctic shallow water sediments [89]. In silico analyses of the predicted amino acid sequence of PET27 revealed a canonical α/β-fold, a conserved catalytic triad (Ser153, Asp198, His230), as well as structural features specific to type IIb PET hydrolases, namely, a three amino acid extension in the loop connecting β8 and α6, and Trp and Ser residues at positions corresponding to Trp159 and Ser238 in *Is*PETase, respectively. However, unlike typical type II hydrolases, which generally possess two disulfide bridges, PET27 exhibited only one, near the C-terminus (Cys262-Cys285), a feature more commonly associated with type I enzymes [72]. Consequently, PET27 cannot be clearly classified into either of the two PET hydrolase types proposed by Joo et al. [74]. Moreover, PET27 displayed a structural feature specific to Bacteroidetes representatives that were not observed in the other characterized PET hydrolases, namely, a PorC-like domain at the C-terminus of the protein. This element, which is part of the type IX secretion system (T9SS), suggested that PET27 is an exoenzyme [72]. The purified recombinant enzyme partially hydrolyzed an amorphous PET foil at 30 °C, releasing 872 µM TPA after a 5-day incubation period (Table 1). This level of product release was 4.7-fold lower than that determined for *Is*PETase under the same reaction conditions [72]. However, when compared to other marine-derived PET hydrolases also assayed on amorphous PET films at 25–30 °C, specifically, Mors1 and PE-H, PET27 demonstrated a higher amount of hydrolysis product release (Table 1).

The PET46 encoding gene was identified in the metagenome-assembled genome of the uncultured *Candidatus* Bathyarchaeota archaeon B1_G2 [73], a strain discovered in deep-sea hydrothermal vent sediments from the Guaymas Basin, Mexico [90]. To the best of our knowledge, PET46 is the only PET-degrading enzyme derived from an archaeon to date. For crystallization and biochemical characterization, the PET46 encoding gene was cloned and expressed in *E*. *coli* [73]. The crystal structure of the purified protein, resolved at 1.71 Å resolution, revealed high levels of similarity to feruloyl/ferulic acid esterases (FAEs; EC 3.1.1.73), which are enzymes that degrade hemicellulose and lignin to ferulic acid and other hydroxycinnamic acids. Specifically, PET46 exhibited an α/β-fold with a core domain composed of an eight-stranded β-sheet flanked by seven α-helices and a lid domain consisting of three α-helices and two anti-parallel β-strands (Figure 6A). The 45-amino-acid-long lid domain (Leu141-Val186) identified in PET46 was not reported in other PET-degrading hydrolases but is a common trait of FAEs, albeit of varying length. Functionally, this structure was essential for the enzymatic activity of PET46, likely due to improved PET substrate accommodation in the active site. Additional structural particularities were observed around the active site, notably in the two loops connecting β4 and α3 (loop 1) and β10 and α10 (loop 2). These loops exhibited high similarity to their homologous structures in FAEs (Figure 6B) while showing significant divergence from those in type I and II PET-degrading enzymes. PET46 possessed a conserved catalytic triad (Ser115, Asp206, His238), consistent with other PET hydrolases. However, it lacked the one or two disulfide bridges that characterize most PET-degrading enzymes reported to date [73].

PET46′s melting temperature of 84.5 °C was almost identical to that of LCC, one of the best-performing thermostable PET-degrading enzymes [32]. This high T_m_ suggested that PET46 is well adapted to function at the elevated temperatures characteristic of the deep-sea hydrothermal vent environment from which it was isolated. Indeed, the enzyme’s activity, determined using *para*-nitrophenyl-decanoate as a substrate, was optimal at 70 °C. The enzyme retained more than 60% of its optimal activity after eight-day incubation at 60 °C but lost almost 80% after two days at 70 °C. PET46 functioned optimally at pH 7–8 while retaining 50% of its maximal activity at pH 5. Moreover, the enzyme remained stable in the presence of various organic solvents and metal ions, some of which had a promoting effect on its activity [73].

The plastic-depolymerizing ability of PET46 was assayed on both amorphous PET foil and semi-crystalline PET powder with over 40% crystallinity. However, while no enzymatic activity was observed on the amorphous substrate, after incubation for three days at 60 °C with the semi-crystalline powder, the enzyme released 1624 µM of aromatic products (Table 1). The main hydrolysis product was TPA (99.1%), followed by MHET (0.88%) and BHET (0.02%). The preference for high-crystallinity substrates observed in PET46, also reported for *Mt*Cut [69] and Enzyme 606 [43], may be attributed to enhanced enzyme–polymer interactions in regions of high-crystallinity, where monomers adopt a more linear arrangement [43]. When the PET-depolymerizing efficacy of PET46 was compared to that of *Is*PETase and LCC at their optimal temperatures, it was found that PET46 produced a similar amount of hydrolysis products as *Is*PETase at 30 °C but a 2.3-fold lower amount than LCC at 50 °C [73]. Moreover, although difficult to accurately compare due to the use of different PET substrates, PET46′s level of product release was similar to that of the marine-derived PET-degrading enzymes CM^A266C^ [63], Ple629 [65], and PET6 and its chimeric variants [64] at temperatures between 30 and 50 °C (Table 1).

In conclusion, the majority of characterized marine-derived PET hydrolases, with the exception of PET46, exhibited optimal catalytic activity at relatively low temperatures. This characteristic, however, represents a challenge for their successful implementation in PET recycling technologies, which require temperatures near PET’s glass transition point for efficient depolymerization [47]. Similar to *Is*PETase, one of the most extensively studied PET hydrolases active at moderate temperatures [21,91], marine-derived enzymes could be subjected to mutation strategies to enhance their activity and thermostability. Until significant progress is achieved in this area, the engineered variants of the thermophilic enzymes LCC [32,92,93] and BhrPETase [94] remain the most effective PET hydrolases and the most promising candidates for enzymatic PET recycling [47]. Nevertheless, PET-degrading enzymes active at moderate temperatures, including those of marine origin, may find application in bioremediation strategies aimed at decomposing micro- and nano-plastics in oceanic sediments [21] and municipal wastewater [47]. In such potential applications, the enzymes’ adaptations to moderate temperatures and high salinity could prove advantageous.

## 3. Polylactic Acid (PLA) Depolymerization

### 3.1. About PLA and Its Biodegradation

Polylactic acid (PLA) is an aliphatic polyester composed of repeating units of 2-hydroxypropanoic acid, commonly known as lactic acid. The monomeric units are primarily produced through microbial fermentation, utilizing lactic acid bacteria and renewable carbohydrate sources, such as corn, sugarcane, or potato. Due to its favorable properties, including biodegradability, biocompatibility, and thermoplastic behavior, PLA has gained considerable attention in various applications, ranging from packaging materials to biomedical devices [4].

Although PLA is generally promoted as a compostable material, its biodegradability is significantly influenced by various intrinsic and extrinsic factors, such as molecular weight, crystallinity, purity, and exposure to abiotic factors (e.g., UV radiations, temperature, moisture, pH), as well as microorganisms with metabolic pathways and enzymes capable of depolymerizing PLA and utilizing it as a carbon source [95]. PLA can persist for decades without substantial degradation in environments, such as seawater, soil, landfills, and home composting systems, where temperatures typically remain below 37 °C. In contrast, under the controlled conditions of industrial composting facilities, where temperatures exceed 60 °C, PLA decomposes relatively rapidly, with significant degradation occurring within approximately 180 days [96].

Despite its biodegradability potential, PLA faces significant challenges in widespread adoption for industrial composting or recycling. The current infrastructure lacks adequate collection systems for biodegradable plastics and sufficient capacity to sort PLA products at their end-of-life. Economic factors also play a crucial role, particularly the lower cost of producing biological monomers through fermentation than recycling PLA. Given these major limitations in industrial composting and recycling, coupled with the fact that lactic acid is a non-toxic monomer that does not negatively impact microbial soil populations or global health, it has become evident that the most promising strategy for PLA disposal is developing methods to enhance its degradation rate in natural environments or home composting systems [4]. For this purpose, microbial PLA-degrading enzymes play an essential role, as these biocatalysts could be embedded in the PLA matrix, thereby creating a self-degrading plastic under specific composting conditions [97,98,99].

The enzymatic hydrolysis of PLA was first reported in 1981 using commercial proteases: proteinase K, pronase, and bromelain [100]. Later, in 2001, the first non-commercial PLA-degrading enzyme was purified from the bacterial strain *Amycolatopsis* sp. 41 [101]. Since then, numerous PLA-depolymerases of bacterial and fungal origin have been reported in the literature, as reviewed recently [4,102]. However, amino acid sequences and detailed biochemical characterization have been provided for only a few enzymes. Indeed, as of August 2024, the PAZy database [62] lists 38 enzymes with PLA-degrading activity, some of which (e.g., Proteinase K, Subtilisin) are available as commercial products. These are generally serine hydrolases from classes EC 3.1 (esterases, lipases, cutinases) and EC 3.4 (proteases) which cleave the PLA ester bonds through a nucleophilic attack by the catalytic serine’s oxygen on the carbonyl carbon atom present in the ester bond [4].

The majority of PLA hydrolases characterized to date originate from soil and compost microorganisms, including *Actinomadura keratinilytica*, *Amycolatopsis* sp., *Lederbergia lenta*, and *Thermobifida cellulosilytica*. These enzymes typically exhibit optimal activity and stability at temperatures exceeding 50 °C (Appendix A) [62]. A smaller proportion of the characterized PLA-degrading enzymes derive from microorganisms inhabiting diverse environments, such as seawater, anaerobic granular sludge, cereals, and lake sediments. These enzymes often exhibit distinctive catalytic properties, reflecting their unique ecological niches (Appendix A). The next section will focus on PLA hydrolases derived from marine environments and their potential utility in developing innovative methods for PLA biodegradation.

### 3.2. PLA-Depolymerizing Marine Enzymes

Of the 38 PLA-degrading hydrolases listed in the PAZy database [62] as of August 2024, eight (excepting engineered variants) originate from marine sources, either cultured microorganisms or environmental metagenomic datasets (Table 2). These enzymes are carboxylesterases (EC 3.1.1.1) that typically depolymerize PLA under moderate temperatures (30–35 °C) and alkaline pH conditions, often exhibiting tolerance to high salinity. However, only two of these enzymes, ABO2449 and RPA1511, have been characterized in greater detail, including analyses of their three-dimensional structures, site-directed mutagenesis studies, and quantitative enzymatic assays.

The PLA-degrading carboxylesterase ABO2449 was derived from the genome of *Alcanivorax borkumensis* SK2 [103], a bacterial strain isolated from seawater and sediment samples collected from the North Sea [107]. The purified recombinant enzyme demonstrated depolymerization activity against both emulsified and solid poly(D,L-lactic acid) (PDLLA) substrates with molecular weights ranging from 2 kDa to 90 kDa. However, the enzyme was inactive against the enantiopure poly(L-lactic acid) and poly(D-lactic acid), likely due to the higher degree of crystallinity of these polymers. After a 36 h incubation of the enzyme with PDLLA powder at 35 °C, pH 8.0, and 0.1% (*w*/*v*) Plysurf A210G, over 90% substrate conversion to lactic acid monomers and oligomers (*n* = 2–13) was achieved (Table 2). While the presence of the surfactant Plysurf did not stimulate enzyme activity, it was essential for achieving such a high PLA depolymerization rate, suggesting that it may facilitate enzyme binding to the solid substrate. The enzyme exhibited optimal esterase activity between 30 and 37 °C, consistent with its relatively low temperature of aggregation (T_agg_) (Table 2). Moreover, it was active within a pH range of from 7.0 to 11.0 (optimally at pH 9.5–10.0) and tolerated up to 3 M of NaCl, although it performed best in the absence of salts. Computational analysis of the amino acid sequence and predicted three-dimensional structure of ABO2449 permitted its classification within the α/β-hydrolase enzyme superfamily, more specifically, to family V of lipolytic enzymes. This classification was consistent with RPA1511 from *Rhodopseudomonas palustris*, but contrasted with other PLA-depolymerizing enzymes (e.g., PlaM4, PlaM5, PlaM7, PlaM8), which were associated with esterase family I [103]. Despite the enzyme’s relatively high PLA-depolymerizing activity, the crystal structure of the protein has not yet been solved, nor have improved engineered variants been developed.

The gene encoding the PLA esterase RPA1511 was identified in the same study as ABO2449 but in the genome of *Rhodopseudomonas palustris* CGA009 [103]. This bacterial species has been found in diverse environments, including swine waste lagoons, earthworm droppings, pond water, and marine coastal sediments [108]. Similar to ABO2449, the purified recombinant RPA1511 exhibited depolymerization activity against different PLA substrates with repetitive D,L units, but not against enantiopure polymers. The enzyme degraded nearly 40% of a solid PDLLA substrate within a 36 h incubation period at 35 °C and pH 8.0 (Table 2). This conversion efficiency of PLA to lactic acid monomers and oligomers was 2.3 times lower than that of ABO2449 under the same reaction conditions. However, this assay was not conducted under the enzyme’s optimal temperature. Indeed, RPA1511 performed best at 55 °C, consistent with its T_agg_ of 70.8 °C. Moreover, it was active within a pH range of from 7.0 to 11.0 (with optimal activity at pH 10.0) and tolerated up to 4 M of NaCl (with optimal activity at 0.5 M). The crystal structure of the enzyme, solved at 2.2 Å resolution, revealed an α/β-hydrolase fold with a core domain composed of eight β-strands surrounded by seven α-helices and a lid domain consisting of five α-helices. The U-shaped lid domain is positioned above the serine residue of the catalytic triad (Ser114, Asp242, His270), leaving the active site cleft open and accessible to the solvent (Figure 7A). A polyethylene glycol molecule, which could mimic PLA, was identified in the active site cleft (Figure 7B–D), and its position suggested that the enzyme can perform both endo- and exoesterase cleavage. Furthermore, site-directed mutagenesis experiments revealed several residues crucial (i.e., Thr48, Gln172, Arg181, Leu212, Met215, Trp218, Leu220, and Lys252) or beneficial (i.e., V202A) for PLA hydrolysis, likely due to their involvement in substrate binding [103].

A recently published study [104] employed protein stability prediction tools and site-directed mutagenesis to computationally redesign RPA1511 and obtain enzyme variants with improved PLA depolymerization activity and enhanced thermal stability. Among the various mutant enzymes created, a variant (designated R5) with five mutations (S153L/I245V/R276D/S128Y/V202W) demonstrated an 8 °C increase in T_m_ and superior PLA depolymerization efficiency compared to the wild-type enzyme. The R5 variant degraded 85% of a solid PDLLA substrate within 72 h under optimal reaction conditions (65 °C, pH 9.0, and 10 g⋅L^−1^ PLA substrate). In contrast, the wild-type enzyme degraded nearly 60% PDLLA within 72 h at its optimal temperature (55 °C) and pH 8.0 (Table 2). Based on molecular docking simulations, it has been suggested that the V202W mutation-induced structural changes in the substrate-binding region leading to improved enzyme-substrate contact and enhanced PLA depolymerization activity. The other four mutations, located distal to the active site, have been predicted to increase R5′s rigidity and thermal stability by forming new hydrogen bonds or strengthening hydrophobic interactions [104].

The other six marine-derived esterases listed in Table 2 have been only partially characterized to date in terms of their PLA-degrading activity [42,105,106]. The purified recombinant enzymes were qualitatively tested using agarose-based screenings and demonstrated the ability to degrade an emulsified PDLLA substrate with a low molecular weight (2 kDa). These enzymes showed relatively low T_agg_ (<50 °C), suggesting the adaptability to function under moderate temperatures. Moreover, three (i.e., ABO_1197, ABO_1251, and MGS0010) exhibited remarkable salt tolerances of up to 3.5 M NaCl [42]. However, detailed quantitative assays using various types of PLA substrates are necessary to fully understand the efficiency of these enzymes in PLA degradation.

## 4. Depolymerization of Other Plastics by Marine-Derived Enzymes

Although plastic polymers other than PET and PLA constitute approximately 93% of the annual global plastic production [3], relatively few enzymes capable of partially degrading these materials have been characterized. As of August 2024, the PAZy database lists 101 enzymes active on polyurethane (30), polyethylene (2), polyamide (21), polyhydroxyalkanoates (16), polybutylene adipate terephthalate (17), polybutylene succinate (4), and natural rubber (11) [62]. These enzymes are predominantly derived from microorganisms inhabiting terrestrial environments, such as soil, sludge, and compost (Appendix A), while only four (i.e., phaZ_Pst_ [109]), Ple628 [65,66], Ple629 [65,66], and PA4-degrading enzyme [110]) are of marine origin. Notably, no enzymes have yet been identified and characterized for other common polymers, such as polypropylene, polyvinyl chloride, and polystyrene [62], although various microbial strains and consortia have been reported to possess degradative potential against these materials [4]. Given the scope of the present work and the fact that the most potent plastic-degrading enzymes have been extensively discussed in a recent review [4], this short section will focus on the four characterized marine-derived enzymes active on biodegradable plastic polymers.

The phaZ_Pst_ enzyme, isolated from the marine strain *Pseudomonas stutzeri* YM1006, has been shown to depolymerize polyhydroxybutyrate (PHB) [109], a polyhydroxyalkanoate often promoted as a substitute for some fossil-based polymers, such as polypropylene and polyethylene [111]. This enzyme remains stable at temperatures of up to 50 °C within the pH range of 6–12 and degrades PHB into monomers of 3-hydroxybutyric acid (3HB). In contrast, other PHB depolymerases generally release 3HB dimers [109]. Structurally, this difference has been suggested to be attributed to the fact that phaZ_Pst_ possesses two substrate-binding domains and recognizes at least two monomeric units of the substrate [112]. However, other specific characteristics distinguishing this marine enzyme from non-marine PHB depolymerases have not been pointed out.

In addition to degrading PET, Ple628 and Ple629 demonstrated the ability to depolymerize polybutylene adipate terephthalate (PBAT), an aliphatic-aromatic co-polyester composed of terephthalic acid (TPA), adipic acid, and 1,4-butanediol. PBAT is generally promoted as a biodegradable alternative to polyethylene and is primarily used in the manufacture of agricultural mulch films, plastic bags, and paper coatings [65]. When assayed on PBAT at its optimal temperature (30 °C), Ple629 yielded 265.9 µM of terephthalate-butanediol monoester (BT) and 18.9 µM TPA after 72 h incubation. At the same time, Ple628 released less than 50 µM of BT after 144 h [65]. By comparison, the most potent PBAT-depolymerase represented by the engineered cutinase TfCut-DM from *Thermobifida fusca* performed the complete PBAT decomposition in 36 h at 60 °C, releasing approximately 1.7 mM TPA and 2 mM BT [113]. A detailed comparison between the various PBAT-depolymerizing enzymes described to date, including Ple628 and Ple629, was recently provided by Yang and collaborators [114]. Also, some structural and functional characteristics of Ple628 and Ple629 were discussed in Section 2.2.

The first marine-derived enzyme capable of degrading polyamide 4 (PA4) was isolated from *Pseudoalteromonas* sp. Y-5 [110]. PA4 is a biobased polyamide that is expected to replace some widely used fossil-based plastics known as Nylons, such as PA6 and PA6.6 [4]. This enzyme degrades PA4 into gamma-aminobutyric acid oligomers under a range of conditions: temperatures of 5–65 °C (optimal at 55 °C), pH 6–10 (optimal at pH 8), and in the presence of various salt concentrations (up to 1 M NaCl). Computational analysis of the amino acid sequence and predicted three-dimensional structure revealed a β-lactamase structure containing serine as an active center residue. The enzyme comprises three domains typically present in other polymer-degrading enzymes: a substrate-binding domain, a linker domain, and a catalytic domain [110]. However, to better evaluate its potential utility in practical recycling applications, further quantitative analysis of this enzyme’s activity on PA4 is necessary.

## 5. Challenges and Perspectives in Enzymatic Plastic Depolymerization

### 5.1. PET

The field of enzymatic PET depolymerization has witnessed remarkable progress in the last five years. Currently, the most potent PET hydrolases are LCC^ICCG^ [32], LCC-A2 [93], and TurboPETase [94]. These enzymes were developed through saturation or site-directed mutagenesis of the metagenome-derived thermophilic enzymes LCC and BhrPETase. LCC^ICCG^ has demonstrated the ability to depolymerize 90% of post-consumer PET waste (10% crystallinity) within 10 h at 72 °C and pH 8. Due to this remarkable performance, the French company Carbios has announced the industrial use of LCC^ICCG^ for enzymatic recycling starting in 2026 [115]. More recently, laboratory-scale studies have shown that LCC-A2 and TurboPETase even outperform LCC^ICCG^. Recent comprehensive reviews [47,116] provide an up-to-date analysis of these and other wild-type and engineered PET hydrolases, offering valuable insights into the current state of the field.

Two major factors constrain the economic viability of industrial-scale enzymatic PET recycling. The first is the necessity for extensive pretreatment of PET waste, including extrusion and micronization [32]. These processes are essential to amorphize the polymer structure and significantly increase the surface area available for enzymatic attack, thereby enhancing degradation efficiency. The second factor is the requirement for continuous pH regulation during the enzymatic process. As PET depolymerizes, it releases terephthalic acid, causing a progressive decrease in pH. To maintain optimal enzyme activity, the addition of a strong base is necessary to neutralize this acidity. Both factors contribute substantially to operational costs and process complexity, presenting significant hurdles to the widespread adoption of this promising recycling technology [4]. To address these challenges, future research could focus on developing PET hydrolases that work more effectively on crystalline PET than on amorphous substrates. Examples of such enzymes include the marine-derived PET46 [73], *Mt*Cut [69], and Enzyme 606 [43]. Understanding the molecular mechanisms behind these enzymes could provide insights into the mutations needed to enhance the catalytic activity of high-performing PET hydrolases on crystalline PET, potentially eliminating the need for extensive pretreatment. Furthermore, the discovery of new PET hydrolases able to tolerate acidic pH is another interesting research direction because it would minimize the need for soda in pH regulation. A good starting point in this regard is the marine-derived enzyme PET46, which retained 50% of its maximal activity at pH 5 [73].

Although more than 100 PET-degrading enzymes have been characterized to date according to the PAZy database [62], none of the wild-type enzymes are suitable for industrial recycling processes [47]. Thus, upcoming research could focus on applying protein engineering methods (reviewed by Yao et al. [116]) to the most promising wild-type enzymes (e.g., PET46 [73]). The search for novel protein scaffolds in underexplored extreme environments, such as deep-sea hydrothermal vents, which, in some cases, combine high temperatures with acidic pH [90], remains equally important. In this regard, a review of the strategies used to identify plastic-degrading microorganisms and enzymes has been recently provided by Zhang and collaborators [117].

In addition to optimizing enzymes for enhanced performance and lifetime under harsh physicochemical conditions—including the presence of inhibitory products, contaminants, and substrates with high crystallinities—numerous other key factors may challenge the economic viability of industrial-scale enzymatic PET recycling compared to de novo production or the chemical and mechanical recycling methods. Some of these factors include the cost of enzyme production, the quantity of enzyme used relative to the amount of substrate, and the extent of conversion. A study that thoroughly analyzed these aspects predicted that PET enzymatic recycling could achieve cost parity with terephthalic acid manufacturing while also offering substantial reductions in environmental impacts compared to the production of virgin polyester [118].

### 5.2. PLA

Widespread adoption of PLA for industrial composting or recycling is currently hindered by inadequate collection systems, insufficient sorting capacity, and economic factors favoring biological monomer production over recycling. Given these limitations and the non-toxic nature of lactic acid, one of the most promising PLA disposal strategies is developing methods to enhance its degradation in natural environments or home composting systems using PLA-degrading enzymes embedded in the PLA matrix [97,98,99]. In this field, significant progress has been made recently by Carbiolice, a subsidiary of the French company Carbios, which has developed an enzymatic product (Carbios Active) that can be successfully embedded in PLA products during their manufacture [119]. This product which is based on an engineered hyperthermostable serine proteinase (designated ProteinT*^FLTIER^*) from *Thermus* sp. Rt4lA triggers the PLA biodegradation under domestic or industrial composting conditions [99]. Indeed, ‘enzymated’ PLA materials completely disintegrated between 20 and 24 weeks of incubation under home-composting conditions (28 °C and 55% relative humidity) and within 50 days under industrial-composing conditions (58 °C and 55% relative humidity) [99].

While this progress in PLA biodegradation demonstrates the high potential for PLA disposal through enzymatic depolymerization, we suppose not many PLA manufacturers would voluntarily choose to include a potentially costly enzymatic product in their production lines without a legislative mandate. Thus, this technology may not achieve its maximal potential without the involvement and support of government authorities to incentivize or require its adoption.

Furthermore, expanding the repertoire of PLA depolymerases could be beneficial in designing new processes for disposing of PLA. Considering that only a limited number of PLA hydrolases have been thoroughly characterized to date, a significant opportunity exists in the purification and characterization of enzymes from the numerous microbial strains reported in the literature as possessing PLA-degrading activity [4].

Current marine PLA depolymerases are primarily effective on PLA substrates with repetitive D,L units, and low molecular weights (2–18 kDa). Future research should focus on bioprospecting marine habitats for novel microbial enzymes capable of depolymerizing high-molecular-weight poly(L-lactic acid). This endeavor could significantly enhance the applicability of these enzymes in PLA composting or recycling processes, where high-molecular-weight polymers (20–300 kDa) comprising over 95% L-units are common [4].

Plastic-degrading enzymes have demonstrated promiscuous abilities to bind and hydrolyze various natural and synthetic polymers [4]. An intriguing future direction is to test the ability of the over 100 known PET hydrolases to degrade other types of plastics, including PLA. This cross-polymer investigation could reveal unexpected catalytic versatilities and potentially lead to the development of multi-substrate plastic-degrading enzymes, which would be highly valuable for mixed plastic waste recycling.

### 5.3. Other Plastics

Although few enzymes active on other plastics than PET and PLA have been thoroughly characterized to date, numerous recent studies have reported the identification of marine microorganisms capable of degrading polyethylene [120,121], polypropylene [122], polyurethane [123], polystyrene [124,125], polyvinyl chloride [126], polybutylene adipate terephthalate [127], polybutylene succinate [128], and polyhydroxyalkanoates [129]. Table 3 highlights some of the most promising sources of enzymes capable of degrading these plastics. Given these findings, future studies are expected to focus on characterizing the enzymes produced by these microorganisms, thereby expanding the repertoire of known plastic-degrading enzymes.

To develop effective enzymatic recycling strategies for the most recalcitrant plastic polymers, there is a compelling need to identify or engineer robust biocatalysts capable of withstanding harsh physicochemical conditions, such as high temperatures and acidic pH. In this regard, expanding the search for enzymes in underexplored extreme environments, particularly deep-sea hydrothermal vents, may prove highly advantageous. These unique ecosystems often combine high temperatures and acidic pH, potentially harboring microorganisms that have evolved enzymes with enhanced stability and activity under industrial conditions. By tapping into these extreme marine environments, researchers may uncover biocatalysts that not only expand our understanding of enzyme diversity but also offer practical solutions for addressing the global challenge of plastic pollution.

## 6. Conclusions

Plastic pollution represents one of the most pressing environmental challenges today, with vast amounts of waste accumulating in terrestrial and marine ecosystems, threatening wildlife and human health. Addressing this issue requires innovative recycling technologies that align with the principles of a circular economy. By adopting such approaches, we can reduce plastic waste, decrease reliance on fossil resources, and lower greenhouse gas emissions, thereby promoting a more sustainable model for the planet.

Enzymatic catalysis of plastics has emerged as a promising recycling solution to overcome the limitations of conventional mechanical and chemical methods, which require harsh physicochemical conditions such as high temperature, high pressure, and toxic chemical catalysts, leading to large energy demands and hazardous waste generation. To date, over 200 enzymes have been reported to degrade various synthetic polymers, particularly PET and PLA. Among them, the most potent were derived from terrestrial thermophilic microorganisms due to the need for most plastics to be efficiently degraded at high temperatures (>65 °C), close to their glass transition point. However, evidence regarding plastic-degrading enzymes sourced from marine microorganisms is rapidly growing, and this review focuses on their potential in advancing recycling technologies.

Marine-derived PET hydrolases, predominantly sourced from cold environments, such as Antarctic seawater and deep-sea sediments, exhibit optimal activity at relatively low temperatures ranging from 25 to 50 °C. An exception is PET46, which originates from hydrothermal vent sediments and demonstrates stability at elevated temperatures. These enzymes typically function optimally in slightly alkaline pH conditions and often exhibit tolerance to high salinities. However, their low-temperature activity presents challenges for the recycling of PET, a process that generally necessitates higher temperatures to be efficient. To address these limitations, there is significant potential for the engineering of these enzymes to improve their thermostability. Future research could also focus on developing PET hydrolases that effectively target crystalline PET. Marine-derived enzymes such as PET46, MtCut, and Enzyme 606 show promise in this regard. Beyond recycling applications, marine enzymes may find utility in the bioremediation of micro- and nano-plastics found in oceanic sediments and wastewater, where their tolerance to moderate temperatures and salinity could prove advantageous.

In addition to PET hydrolases, several marine-derived enzymes have been characterized for their activity against PLA and other biodegradable plastics. Current marine PLA depolymerases are primarily effective on PLA substrates with repetitive D,L units and low molecular weights. Consequently, their applicability in PLA composting or recycling processes is limited, as PLA waste products typically consist of high-molecular-weight polymers that contain over 95% L-units. Therefore, future research should focus on identifying more effective biocatalysts capable of degrading a broader range of synthetic polymers beyond PET.

In conclusion, while the currently characterized plastic-degrading enzymes from marine microorganisms may not surpass the potency of the most effective biocatalysts derived from terrestrial thermophilic microorganisms, they significantly broaden the range of available enzymes. This expanded repertoire presents important opportunities for further protein engineering efforts, paving the way for developing more efficient enzymatic recycling strategies for plastics.

## Figures and Tables

**Figure 1 marinedrugs-22-00441-f001:**
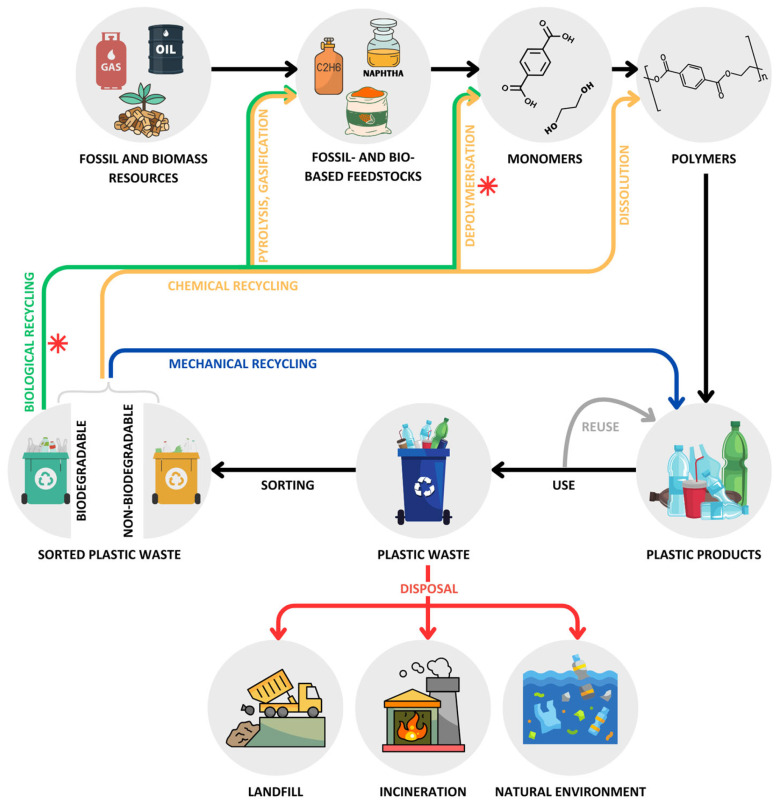
Plastic life cycle: from production using fossil or biomass resources to disposal or recycling through mechanical (blue line), chemical (yellow line), and biological (green line) methods. Recycling processes involving enzymes and/or microorganisms are indicated with a red asterisk (*).

**Figure 2 marinedrugs-22-00441-f002:**
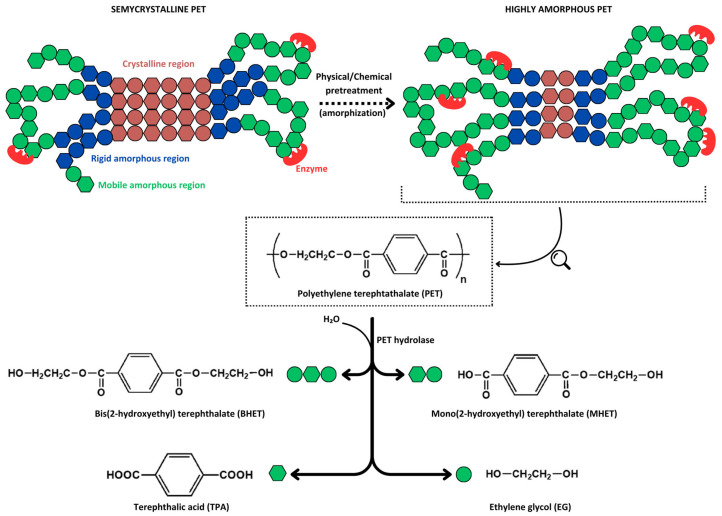
Schematic representation of the enzymatic hydrolysis of semicrystalline and highly amorphous PET. Hydrolases preferentially bind to the mobile amorphous regions, breaking them down into simpler molecules: TA, EG, MHET, and BHET.

**Figure 3 marinedrugs-22-00441-f003:**
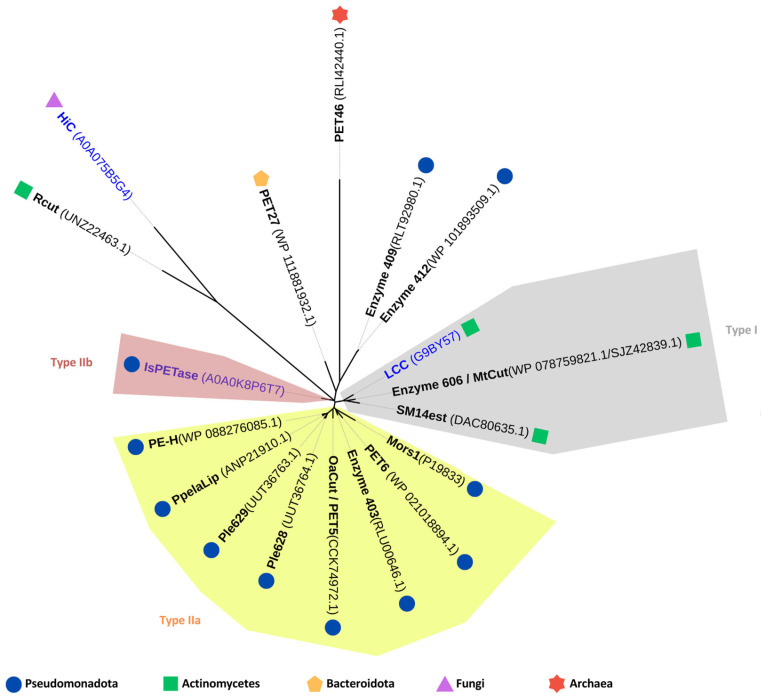
Unrooted phylogenetic tree of 15 PET hydrolases derived from marine microorganisms, along with LCC, *Is*PETase, and HiC. The tree illustrates three distinct enzyme clusters: type I (grey background), type IIa (yellow background), and type IIb (reddish background), as per the classification proposed by Joo et al. [74]. Unclassified enzymes are shown on a white background. Symbols in the legend indicate the taxonomic origin of each PET hydrolase. Amino acid sequences, with accession numbers in parentheses, were obtained from GenBank or UniProtKB. Sequence alignment was performed using ClustalW version 2.0. The phylogenetic tree was constructed using MEGA X 10.0 software [75] with the maximum likelihood method, employing the LG + F model and 100 bootstrap replications.

**Figure 4 marinedrugs-22-00441-f004:**
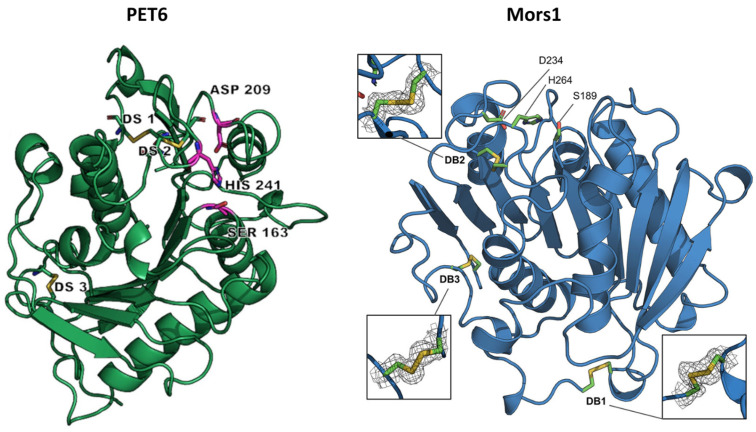
Cartoon representations of the three-dimensional structures of PET6 and Mors1. Disulfide bonds are labeled DS1–3 in PET6 and DB1–3 in Mors1. Insets for Mors1 show close-ups of the three DBs, with black mesh representing the electron density of each DB. The catalytic residues Ser-Asp-His are shown in both representations. Reproduced with permission from ref [64] (PET6) and ref [63] (Mors1).

**Figure 5 marinedrugs-22-00441-f005:**
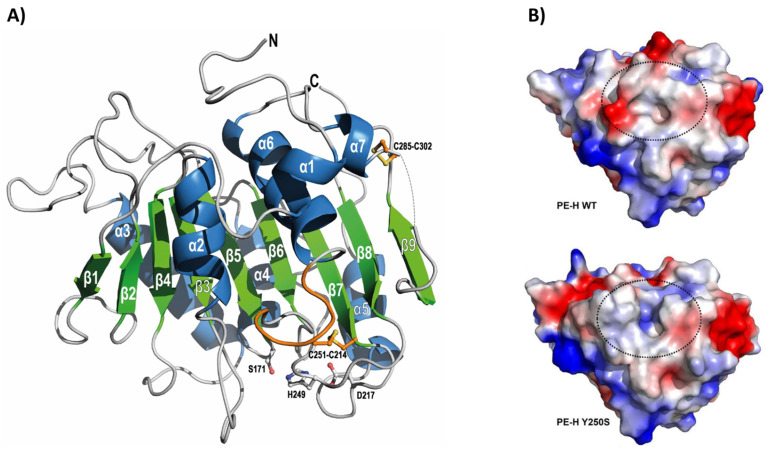
Crystal structure of PE-H. (**A**) Cartoon representation of the three-dimensional structure. The extended loop region and the Cys residues forming disulfide bonds are highlighted in orange. Residues of the catalytic triad are shown as gray ball-and-stick models with labels. (**B**) Surface representation of the wild-type (WT) PE-H and the Y250S mutant variant. The electrostatic surface is color-coded: blue for positive charge and red for negative charge. The active site cleft is indicated by a dashed line. Adapted from ref [68].

**Figure 6 marinedrugs-22-00441-f006:**
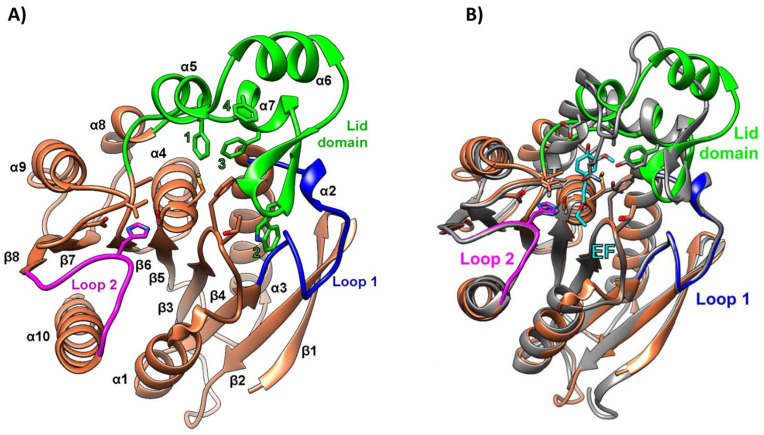
Crystal structure of PET46. (**A**) Cartoon representation of the three-dimensional structure, with highlighted lid domain, loop 1, and loop 2. The four aromatic residues of the lid domain (1–4, bright green) and the residues involved in catalytic activity and substrate binding are shown as stick models. (**B**) Cartoon representation of PET46′s three-dimensional structure (coral orange) overlaid with the structure of the cinnamoyl esterase LJ0536 S106A mutant from *Lactobacillus johnsonii* (dark gray) in complex with ethylferulate (EF, cyan). Adapted from ref [73].

**Figure 7 marinedrugs-22-00441-f007:**
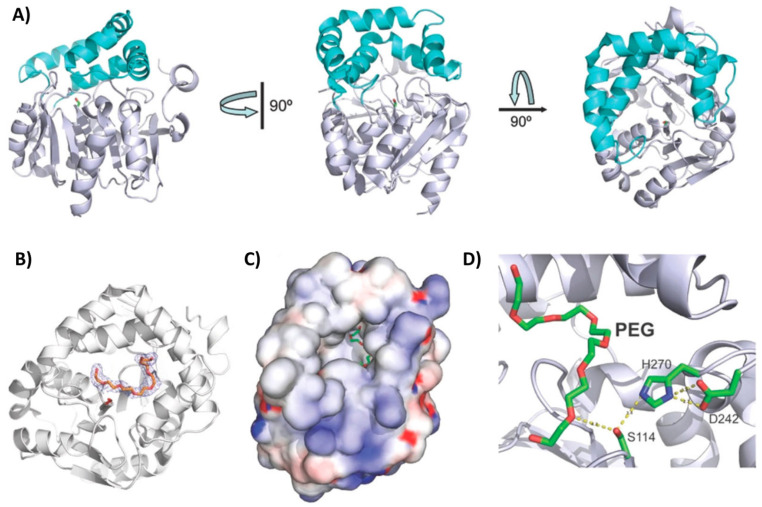
Crystal structure of RPA1511. (**A**) Cartoon representation of the three-dimensional structure, with the core domain shown in gray and the lid domain colored in cyan. (**B**) Protomer with bound PEG 3350 (dodecaethylene glycol, shown as sticks). (**C**) Surface representation of the protomer revealing the active site cleft with bound PEG 3350 (shown as green sticks). The electrostatic surface is color-coded: blue for positive charge, red for negative charge, and white for neutral. (**D**) Close-up view of the PEG 3350 molecule bound close to the catalytic triad (shown as sticks). Adapted with permission from ref [103]. Copyright 2016 American Chemical Society.

**Table 1 marinedrugs-22-00441-t001:** Reported PET-depolymerizing enzymes from marine microorganisms.

Name	Source	Isolation Habitat	T_m_ (°C)	PET Hydrolysis and Reaction Conditions	Ref.
OaCut (PET5)	*Oleispira antarctica*	Antarctic seawater	40.4 ^1^	0.4 µM enzyme produced 0.4% weight loss of amorphous PET films after 6 days at 25 °C, pH 8.0	[39,41]
Mors1 (wild type)	*Moraxella* sp.	Antarctic seawater	52.0 ^1^	0.4 µM enzyme produced 1.98% weight loss of amorphous PET films after 6 days at 25 °C, pH 8.0; ~0.26 mM hydrolysis products were released after 24 h under the same reaction conditions	[39]
CM^A266C^ (chimeric Mors1 variant)	*Moraxella* sp.	Antarctic seawater	55.7 ^1^	0.7 µM enzyme produced 2.5% weight loss of amorphous PET films (~1.5 mM hydrolysis products) after 24 h at 45 °C, pH 8.0	[63]
PET6 (wild type)	*Vibrio gazogenes*	Marine mud	49.8–57.7 ^2,3^	2 µM enzyme produced 1.1 mM hydrolysis products from post-consumer PET (~10% crystallinity) after 18 h at 50 °C, pH 8.5 and 1 M NaCl	[41,64]
PET6-VSTA (PET6 variant)	*Vibrio gazogenes*	Marine mud	48.8–56.7 ^2,3^	2 µM enzyme produced ~1.75 mM hydrolysis products from post-consumer PET (~10% crystallinity) after 18 h at 50 °C, pH 8.5 and 1 M NaCl	[64]
PET6-ExLoop (PET6 variant)	*Vibrio gazogenes*	Marine mud	n.d.	2 µM enzyme produced ~2 mM hydrolysis products from post-consumer PET (~10% crystallinity) after 18 h at 50 °C, pH 8.5 and 1 M NaCl	[64]
Ple628	*Marinobacter* sp.	Marine sediment	41.4 ^1^ 47.1 ^2^	0.6 µM enzyme produced 0.062 mM hydrolysis products from PET nanoparticles after 72 h at 30 °C, pH 7.4	[65,66]
Ple629	*Marinobacter* sp.	Marine sediment	38.1 ^1^ 43.2 ^2^	0.6 µM enzyme produced 1.5 mM hydrolysis products from PET nanoparticles after 72 h at 30 °C, pH 7.4	[65,66]
PpelaLip	*Halopseudomonas* *pelagia*	Antarctic algae	n.d.	1 µM enzyme produced ~10 mmol TPA/mol polyester (equivalent to ~17 µM TPA) from amorphous (<1% crystallinity) in-house synthesized PET analog after 7 days at 28 °C, pH 7.0	[67]
PE-H	*Halopseudomonas* *aestusnigri*	Intertidal sand	50.8 ^1^	0.5 µM enzyme produced 4.2 mg/L MHET (equivalent to 20 µM MHET) from amorphous PET film after 48 h at 30 °C, pH 7.4	[68]
PE-H^Y250S^ (PE-H variant)	*Halopseudomonas* *aestusnigri*	Intertidal sand	49.8 ^1^	0.5 µM enzyme produced 5.4 mg/L MHET (equivalent to 26 µM MHET) from amorphous PET film, and 0.12 mg/L MHET (equivalent to 0.57 µM) from semicrystalline PET after 48 h at 30 °C, pH 7.4	[68]
Enzyme 403	*Ketobacter* sp.	Deep-sea metagenome	n.d.	0.7 mg enzyme/g PET (equivalent to ~0.65 µM enzyme) produced 1.4–1.7 mg/L aromatic products (equivalent to ~9 µM) from amorphous PET film after 96 h at 70 °C, pH 6.0–9.0	[43]
Enzyme 409	*Ketobacter* sp.	Deep-sea metagenome	n.d.	0.7 mg enzyme/g PET (~0.69 µM enzyme) produced 9.8 mg/L aromatic products (equivalent to ~50 µM) from amorphous PET film after 96 h at 60 °C, pH 9.0	[43]
Enzyme 412	*Ketobacter* sp.	Surface seawater	n.d.	0.7 mg enzyme/g PET (~0.65 µM enzyme) produced 2.2 mg/L aromatic products (equivalent to ~11 µM) from amorphous PET film after 96 h at 60 °C, pH 6.0	[43]
Enzyme 606 (*Mt*Cut)	*Marinactinospora* *thermotolerans*	Deep-sea sediment	53.9 ^2^	0.7 mg enzyme/g PET (~0.69 µM enzyme) produced 67 mg/L aromatic products (equivalent to ~345 µM) from amorphous PET film after 96 h at 60 °C, pH 9.0	[43]
			33.0–41.5 ^1,4^	5 mg enzyme/g PET (~0.3 µM enzyme) produced 400 µM aromatic products from PET microparticles (42% crystallinity) after 120 h at 40 °C, pH 8.5	[69]
Rcut	*Rhodococcus* sp.	Antarctic Ross Sea	n.d.	0.026 µM of enzyme produced traces of hydrolysis products from PET film after 24 h at 30 °C, pH 9.0	[70]
SM14est	*Streptomyces* sp.	Marine sponge	55.0 ^1^	0.5 µM of enzyme produced 0.27 mM hydrolysis products from semicrystalline PET powder (>40% crystallinity) after 7 h at 45 °C, pH 8.0 and 0.5 M NaCl	[71]
PET27	*Aequorivita* sp.	Antarctic sediments	n.d.	28.6 mg enzyme/g PET (~26.5 µM enzyme) produced 0.872 mM TPA from amorphous PET film after 120 h at 30 °C, pH 8.0	[72]
PET46	*Candidatus* Bathyarchaeota	Deep-sea hydrothermal vent sediments (metagenome)	84.5 ^1^	3 µM enzyme produced 1.6 mM TPA from semicrystalline PET powder (>40% crystallinity) after 72 h at 60 °C, pH 8.0	[73]

Abbreviations: T_m_ = melting temperature; TPA = terephthalic acid; MHET = mono(2-hydroxyethyl) terephthalate; n.d. = not determined; ^1^ T_m_ values determined by nano differential scanning fluorimetry (nanoDSF); ^2^ T_m_ values determined by differential scanning calorimetry (DSC); ^3^ T_m_ changes with varying salt concentrations; ^4^ T_m_ changes with varying Ca^2+^ concentrations.

**Table 2 marinedrugs-22-00441-t002:** Reported PLA-depolymerizing enzymes from marine microorganisms.

Name	Source	Isolation Habitat	T_m_/T_agg_ (°C)	PLA Hydrolysis and Reaction Conditions	Ref.
ABO2449	*Alcanivorax borkumensis*	Seawater/sediments	T_agg_ = 32.3	4 mg enzyme⋅g PLA^−1^ produced 120 mM lactate (~90% substrate conversion) from PDLLA powder (Mw: 10–18 kDa) after 36 h at 35 °C, pH 8.0, and 0.1% (*w*/*v*) Plysurf A210G	[103]
RPA1511	*Rhodopseudomonas palustris*	Various sources, including marine sediments	T_agg_ = 70.8	4 mg enzyme⋅g PLA^−1^ produced 50 mM lactate (~40% substrate conversion) from PDLLA powder (Mw: 10–18 kDa) after 36 h at 35 °C, pH 8.0	[103]
			T_m_ = 70.1 ^1^	4 mg enzyme⋅g PLA^−1^ produced ~70 mM lactate (~60% substrate conversion) from PDLLA powder (Mw: 10–18 kDa) after 72 h at 55 °C, pH 8.0	[104]
R5 (RPA1511 variant)	*Rhodopseudomonas palustris*	Various sources, including marine sediments	T_m_ = 78.7 ^1^	4 mg enzyme⋅g PLA^−1^ produced 94.5 mM lactate (~85% substrate conversion) from PDLLA powder (Mw: 10–18 kDa) after 72 h at 65 °C, pH 9.0	[104]
MGS0109	Uncultured bacterium	Seawater metagenome	T_agg_ = 48.1	The PLA-degrading activity was confirmed through a qualitative assay on agar plates containing emulsified PDLLA (Mw: 2 kDa) after 24 h at 30 °C, pH 8.0	[42]
MGS0010	Uncultured bacterium	Seawater metagenome	T_agg_ = 46.2	The PLA-degrading activity was confirmed through a qualitative assay on agar plates containing emulsified PDLLA (Mw: 2 kDa) after 24 h at 30 °C, pH 8.0	[42]
MGS0105	Uncultured bacterium	Seawater metagenome	T_agg_ = 46.1	The PLA-degrading activity was confirmed through a qualitative assay on agar plates containing emulsified PDLLA (Mw: 2 kDa) after 24 h at 30 °C, pH 8.0	[42]
ABO_1197	*Alcanivorax borkumensis*	Seawater metagenome	T_agg_ = 47.0	The PLA-degrading activity was confirmed through a qualitative assay on agar plates containing emulsified PDLLA (Mw: 2 kDa) after 24 h at 30 °C, pH 8.0	[42]
ABO_1251	*Alcanivorax borkumensis*	Seawater metagenome	T_agg_ = 45.7	The PLA-degrading activity was confirmed through a qualitative assay on agar plates containing emulsified PDLLA (Mw: 2 kDa) after 24 h at 30 °C, pH 8.0	[42]
MGS0084	Uncultured organism	Tar samples from a sunken shipwreck	n.d.	The PLA-degrading activity was confirmed through a qualitative assay on agar plates containing emulsified PDLLA (Mw: 2 kDa), at 30 °C, pH 8.0	[105,106]

Abbreviations: T_m_ = melting temperature; T_agg_ = aggregation temperature; PDLLA = Poly-D,L-lactic acid; n.d. = not determined; ^1^ T_m_ values determined by differential scanning fluorimetry (DSF).

**Table 3 marinedrugs-22-00441-t003:** Degradation capacity of marine microorganisms for less investigated plastic types.

Type of Plastic	Microorganisms	Isolation Source	Degrading Conditions	Substrate (Degradation Effectiveness)	Ref.
PE	*Alcanivorax borkumensis*	Marine plastic debries	30 °C, 0.05% hexadecane, 80 days	LDPE (3.5% weight loss)	[130]
	*Alcanivorax* sp. *24*	Marine plastic debries	25 °C, 34 days	LDPE (23% MW) and 0.9% overall mass decrease	[131]
	*Exiguobacterium* sp., *Halomonas* sp., *Ochrobactrum* sp.	Seawater	14 days	LDPE (100%)	[132]
	*Alcaligenes faecalis*	Seawater	37 °C, 70 days	UV-treated PE (47.36% weight loss)	[133]
	*Marinobacter* spp., *Bacillus subtilis*	Marine water and sediment	90 days	LDPE (1.46–1.68% weight loss)	[134]
	*Aspergillus flavus*, *A. niger*, *A. fumigatus*, *A. terreus*, *Aspergillus* sp., *Penicillium* sp.	Seawater	28 °C, 30 days	LDPE 16.2–43.3% weight loss)	[135]
	*Alternaria alternata* FB1	Seawater	28 days 120 days	PE film (decrease of crystallinity from 62.79% to 52.02% and of MW by 95%)	[120]
	*Parengyodontium album*	North Pacific plastic debris	^13^C-PE, 9 days	UV-treated PE (0.05%/day to CO2)	[136]
	*Rhodotorula mucilaginosa*	North Sea plastic debris	^13^C-PE, 5 days	UV-treated PE (3.8% yr^−1^ to CO_2_)	[137]
	*Zalerion maritimum*	Marine coastal water	25 °C, 14 days	PE (56.7 ± 2.9% weight loss)	[138]
PP/PS	*Alcanivorax borkumensi* sw2	Mesopelagic seawater (374 m)	10 °C	PP oxidation	[139]
	*Pseudoalteromonas lipolytica*, *Pseudoalteromonas tetraodonis*	Pacific deep sea	15 °C, 80 days	PP film (1.3% and 0.7%) weight loss PS film (3.9% and 2.8%) weight loss	[140]
	*Rhodococcus* sp. *36*, *Bacillus* sp. *27*	Marine sediments	20 °C, 40 days	PP (6.4% and 4% weight loss)	[141]
PUR	*Bacillus velezensis*	Deep sea	37 °C oxidoreductase Oxr-1	Waterborne PUR and PBAT film	[123]
	*Cladosporium halotolerans 6UPA1*	Deep sea	28 °C, 3 days	Impranil PUR (80% weight loss), Esterase, ChLip1 (lipase) and ChCut1 (cutinase),	[142]
PHA	*Bacillus* sp. MH10, *Alteromonas* sp. MH53, *Psychrobacillus* sp. PL87, *Rheinheimera* sp. PL100	Deep sea	10 MPa, 4 °C	PHBH	[143]
	*Pseudomonas* sp., *Alcanivorsax* sp., *Tenacibaculum* sp.	Deep sea	30 MPa, 4 °C	PHB, PHV	[144]
	*Alcaligenes faecalis*	Coastal seawater	55 °C, pH 9.0	PHB	[145]
	*Comamonas testosteroni*	Sea water	pH 9.5–10.0	PHB, PHBV, P3HB4HB	[146]
	*Microbulbifer* sp. *SOL03*	Marine sediments	37 °C, 10 days	PHB (97% weight loss)	[147]
	*Bacillus* sp. *JY14*	Marine sediments	30 °C, 7 days	PHB (98% weight loss) P(3HB4HB) P(3HB3HV)	[148]
	*Candida guilliermondi M-122, Debaryomyces hansenii M-113*	Surface seawater	0.1–20 MPa, 35 °C	PHB (Maximum clearing zone)—3–7 days 0.1 MPa	[149]
	*Rhodosporidium sphaerocarpum M-185*	Deep sea (1494 m)	0.1–30 MPa, 35 °C	PHB (Maximum clearing zone)—3–5 days 0.1 MPa	[149]
	*Aspergillus ustus M-224*	Deep sea (650 m)	0.1–30 MPa, 27 °C	PHB (Vmax = 3U/mL)	[149]

Abbreviations: PHBV = Poly(3-hydroxybutyrate-co-3-hydroxyvalerate; P3HB4HB = Poly(3-hydroxybutyrate-co-4-hydroxybutyrate); PHBH = Poly 3-hydroxybutyrate-co-3-hydroxyhexanoate.

## Data Availability

No new data were created or analyzed in this study. Data sharing is not applicable to this article.

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
