# Peer review of "Plastic-Degrading Enzymes from Marine Microorganisms and Their Potential Value in Recycling Technologies"

_marinedrugs, 2024, doi:10.3390/md22100441_

Round 1

Reviewer 1 Report

Comments and Suggestions for Authors

Dear Authors, please see my comments below

Line 41-42 - The authors should provide a more detailed description of polymer biodegradation, e.g. the stages and minimum requirements to recognize the biodegradation process, with appropriate citations. Some biodegradation processes begin with surface erosion or bulk erosion, particularly in aqueous conditions (closely related to the enzyme's medium). Therefore, connecting this effect with the enzyme degradation stages would strengthen the quality of the article. Reference it with up-date articles (<4 years).

Figure 1 - The illustration of polymers is inadequate; it does not accurately represent a polymer (e.g., it lacks a continuous chain symbol). Please correct it

Figure 1 - Biodegradable polymers can also be chemically and mechanically recycled, but most cases result in down-cycling. The authors should correct the figure and ensure accuracy.

Line 100 - Despite the enzymatic hydrolysis of PET, the degradation of aromatic polyester cannot be recognized as naturally biodegradable, as the authors suggested. For a polymer to be considered biodegradable, it must be broken down by microorganisms under natural conditions (rather than in specific lab settings) within a reasonable timeframe (a few months), displaying markers of biodegradation. This is why I recommend that the authors add more in-depth knowledge about polymer biodegradation and its stages. Doing so will improve the quality of the article and demonstrate a more accurate understanding of the subject.

Figure 2 - The pretreatment step could be better explained. Is it chemical, physical, or both? What types are involved? Display this information clearly.

Figure 2 - While the content may be easily understood by experts, the term "amorphization" is not defined, either in a footnote or in the main text. This should be addressed.

The review alternates between terms like “plastic-degrading enzymes,” “plastic hydrolases,” and “depolymerases” without clear distinctions or definitions. These differences should be clarified, especially considering that the article will also be read by non-experts. A more consistent use of terminology and clear definitions of specific enzyme categories early on would help avoid confusion.

When discussing marine-derived enzymes, it would be helpful to provide a more detailed comparison with land-based or thermophilic organisms. Are there specific features that make these marine organisms superior for certain types of plastic degradation?

Line 673 - The authors should remove references to the specific time of writing and instead use verifiable data and citations. For example, the sentence could be revised as: "As of June 2023, the PlasticDB database lists 753 species of microorganisms capable of breaking down plastics."

Unnecessary temporal references, e.g. "at the time of writing," should be avoided... This phrase appears frequently throughout the article.

Improvements needed: The text discusses the advantages of marine-derived enzymes for industrial recycling but underplays the significant technical challenges in scaling enzymatic recycling,e.g.low yields and high costs compared to traditional chemical recycling methods. The authors should include a section discussing these challenges.

Line 921-942 - The authors describe the need to explore and identify enzymes in harsh environments like deep-sea hydrothermal vents for use in plastic degradation. However, as described in the introduction, the article should also highlight the potential for engineering enzymes to work in milder conditions, where energy waste and the use of harsh chemicals would be minimized. Despite the high technological challenges, the authors could suggest strategies that might serve as a path for future research in this area.

The authors should provide a conclusion section instead of solely focusing on perspectives. This section could also address future directions, which remain vague in the current version. For instance, it could discuss the energy efficiency and environmental impact of enzymatic recycling methods...

Comments on the Quality of English Language

No need.

Author Response

We appreciate the reviewer’s constructive comments and suggestions on our manuscript. We have done our best to address all the feedback and revised the paper accordingly. The changes are highlighted in yellow, and detailed in our responses to each comment below.

Reviewer’s comment: Line 41-42 - The authors should provide a more detailed description of polymer biodegradation, e.g. the stages and minimum requirements to recognize the biodegradation process, with appropriate citations. Some biodegradation processes begin with surface erosion or bulk erosion, particularly in aqueous conditions (closely related to the enzyme's medium). Therefore, connecting this effect with the enzyme degradation stages would strengthen the quality of the article. Reference it with up-date articles (<4 years).

We have added a more detailed description of the stages involved in polymer biodegradation and included two recent citations from the last four years to support this information (lines 92-98). However, we have not included detailed information on the criteria for recognizing polymer biodegradation, as this would overwhelm the introduction with information that falls outside the scope of the manuscript. We believe the additional details on the biodegradation stages enhance the discussion while maintaining the focus of the article.

Reviewer’s comment: Figure 1 - The illustration of polymers is inadequate; it does not accurately represent a polymer (e.g., it lacks a continuous chain symbol). Please correct it

We have revised Figure 1 as suggested, for an accurate representation of a polymer.

Reviewer’s comment: Figure 1 - Biodegradable polymers can also be chemically and mechanically recycled, but most cases result in down-cycling. The authors should correct the figure and ensure accuracy.

We corrected Figure 1 to reflect that biodegradable polymers can be chemically and mechanically recycled.

Reviewer’s comment: Line 100 - Despite the enzymatic hydrolysis of PET, the degradation of aromatic polyester cannot be recognized as naturally biodegradable, as the authors suggested. For a polymer to be considered biodegradable, it must be broken down by microorganisms under natural conditions (rather than in specific lab settings) within a reasonable timeframe (a few months), displaying markers of biodegradation. This is why I recommend that the authors add more in-depth knowledge about polymer biodegradation and its stages. Doing so will improve the quality of the article and demonstrate a more accurate understanding of the subject.

We have revised the manuscript to replace the term "PET biodegradation" with a more precise expression. We now use "enzymatic catalysis of plastic" to emphasize that PET degradation is driven by enzyme activity under controlled conditions, rather than through natural biodegradation. (lines 9; 106-107).

Also, we have added a more detailed description of the stages involved in polymer biodegradation (lines 92-98).

Reviewer’s comment: Figure 2 - The pretreatment step could be better explained. Is it chemical, physical, or both? What types are involved? Display this information clearly.

Indeed, different types of physical and/or chemical pretreatments can be used to increase the amorphous fraction of PET. We have revised Figure 2 accordingly, now stating "Physical/Chemical pretreatment" without specifying a particular method to avoid overwhelming the figure with excessive details. Also, we have added some examples of physical and chemical pretreatments in the manuscript (lines 211-212).

Reviewer’s comment: Figure 2 - While the content may be easily understood by experts, the term "amorphization" is not defined, either in a footnote or in the main text. This should be addressed.

For clarity, we have introduced the term "amorphization" in the main text (line 212).

Reviewer’s comment: The review alternates between terms like “plastic-degrading enzymes,” “plastic hydrolases,” and “depolymerases” without clear distinctions or definitions. These differences should be clarified, especially considering that the article will also be read by non-experts. A more consistent use of terminology and clear definitions of specific enzyme categories early on would help avoid confusion.

To date, no universally accepted term exists for plastic-degrading enzymes. Various names are in use, such as "plastic hydrolytic enzymes", "plastic hydrolases", "plastic depolymerases", or even "PETases". Therefore, the different terms “plastic-degrading enzymes,” “plastic hydrolases,” and “depolymerases” were interchangeably used to avoid repetition and enhance readability. Meanwhile, a clear definition was added in the manuscript (lines 121-122).

Reviewer’s comment: When discussing marine-derived enzymes, it would be helpful to provide a more detailed comparison with land-based or thermophilic organisms. Are there specific features that make these marine organisms superior for certain types of plastic degradation?

Comparison of marine-derived enzymes with land-based or thermophilic organisms was already carried out throughout the manuscript. For instance, we compared efficient marine enzymes such as PET6, Mors1, PET27, and PET46 with well-studied enzymes like IsPETase from land-based bacteria and LCC from thermophilic strains (lines 339-343, 360, 575-576, 627-630). Additionally, in Section 5.1 (lines 865-877), we highlighted specific advantageous features of marine enzymes PET46, MtCut, and Enzyme 606.

However, for a more comprehensive overview, we have consolidated this information in Section 6 (Conclusions).

Reviewer’s comment: Line 673 - The authors should remove references to the specific time of writing and instead use verifiable data and citations. For example, the sentence could be revised as: "As of June 2023, the PlasticDB database lists 753 species of microorganisms capable of breaking down plastics."

Unnecessary temporal references, e.g. "at the time of writing," should be avoided... This phrase appears frequently throughout the article.

This expression was deleted along the manuscript.

Reviewer’s comment: Improvements needed: The text discusses the advantages of marine-derived enzymes for industrial recycling but underplays the significant technical challenges in scaling enzymatic recycling,e.g.low yields and high costs compared to traditional chemical recycling methods. The authors should include a section discussing these challenges.

We have renamed Section 5 as "Challenges and Perspectives in Enzymatic Plastic Depolymerization" and included a paragraph about the major challenges in scaling enzymatic recycling (lines 887-896) among the others already discussed (lines 858-877).

Reviewer’s comment: Line 921-942 - The authors describe the need to explore and identify enzymes in harsh environments like deep-sea hydrothermal vents for use in plastic degradation. However, as described in the introduction, the article should also highlight the potential for engineering enzymes to work in milder conditions, where energy waste and the use of harsh chemicals would be minimized. Despite the high technological challenges, the authors could suggest strategies that might serve as a path for future research in this area.

We highlighted the need to explore extreme marine habitats, such as deep-sea hydrothermal vents, as these environments often combine high temperatures with acidic pH, increasing the likelihood of discovering novel enzymes resistant to such harsh conditions. As outlined in different sections of the manuscript (lines 196-216, 634-638), PET recycling technologies require temperatures near PET's glass transition point (70-80 °C) for efficient depolymerization. At these temperatures, PET chains become mobile, promoting enzyme binding. Additionally, PET hydrolases capable of tolerating acidic pH would reduce the need for soda in pH regulation during recycling, as PET depolymerization releases terephthalic acid, leading to a progressive drop in pH, which inactivates conventional enzymes.

We mentioned in the introduction (lines 80-84) that some emerging recycling technologies (such as enzymatic, photo-, electro-, and microwave-assisted catalysis) may be more environmentally friendly and energy-efficient than conventional chemical recycling processes, which usually require very high temperatures (>180 °C), high pressure (20-40 atm), and toxic chemical catalysts. However, many of these emerging technologies still require temperatures around 70-80 °C. For this reason, we used the expression 'relatively mild conditions' (line 83).

Most wild-type marine enzymes discussed in the manuscript exhibit optimal catalytic activity under mild conditions, which may not be suitable for efficient plastic recycling technologies. However, as mentioned in the manuscript (lines 878-882), future research could focus on applying protein engineering methods to enhance the most promising of these biocatalysts (e.g., PET46 enzyme which is thermostable, relatively resistant to acidic pH, and relatively effective on crystalline PET). Additional perspectives for future research have already been presented in Section 5 (lines 873-877, 913-923).

Reviewer’s comment: The authors should provide a conclusion section instead of solely focusing on perspectives. This section could also address future directions, which remain vague in the current version. For instance, it could discuss the energy efficiency and environmental impact of enzymatic recycling methods...

We have introduced a Conclusion section that provides an overview of the key discussions presented in the manuscript and addresses future research directions. We also addressed the energy efficiency and environmental impact of enzymatic recycling methods, as well as the challenges and opportunities they present compared to traditional recycling processes.

Reviewer 2 Report

Comments and Suggestions for Authors

The authors offer a detailed and extensive review of plastic-degrading enzymes isolated from marine microorganisms.   Their work centers on PET and PLA degradation, as they are the most widely studied enzymatic degradation examples.   I think they should point out some of their findings in the conclusion or final remarks. For example, many isolates have been obtained from cold environments when PET degradation requires high temperatures because of its high Tg, the novel bacteria classes, etc. Besides, even if it is an extended review, a more detailed table of the microbial sources that degrade other polymers would be a significant contribution, as these have not been reviewed systematically (PE, PU, for example).

Author Response

We thank the reviewer’s constructive comments and suggestions on our manuscript. We have done our best to address all the feedback and revised the paper accordingly. The changes are highlighted in yellow, and detailed in our responses to these comments below.

Reviewer’s comment: The authors offer a detailed and extensive review of plastic-degrading enzymes isolated from marine microorganisms. Their work centers on PET and PLA degradation, as they are the most widely studied enzymatic degradation examples. I think they should point out some of their findings in the conclusion or final remarks. For example, many isolates have been obtained from cold environments when PET degradation requires high temperatures because of its high Tg, the novel bacteria classes, etc. Besides, even if it is an extended review, a more detailed table of the microbial sources that degrade other polymers would be a significant contribution, as these have not been reviewed systematically (PE, PU, for example).

We have added a Conclusion section that provides an overview of the key discussions presented in the manuscript.

We have also introduced a new table (Table 3) presenting promising microbial sources of plastic-degrading enzymes.

Reviewer 3 Report

Comments and Suggestions for Authors

The document entitled " Plastic-Degrading Enzymes from Marine Microorganisms and Their Potential Value in Recycling Technologies" deals with the actual state of the art regarding enzymes derived form marine microorganisms capable of degrading plastics, authors present a clear picture of the plastic waste problem and the present solutions along with the advantages of using enzymes to deal with it.

The review includes several aspects of the reported plastic degrading enzymes, including some structural anaylisis.

General remarks

The document is well written and provides a general picture of the known and unknown potential in the plastic-degrading enzymes from marine microorganisms. The plagiarism detected was lower than 3.5% which is adequate.

Comments: 

Please consider the number of Keywords, 3-5 are generally ideal.

Is there any possibility of including an abbreviations table? It would help the reader to consult along the document the use of abbreviations.

The introduction deals mainly with data regarding the production of plastics and a good panorama of the plastic life cycle.

In Lines 75-76 that “(i.e., monomers, oligomers, polymers) that can be used for various practical purposes, including creating value-added products” It would be interesting to address in this section more detail regarding the monomer’s characteristics or potential toxicity or the perspective that those monomers could be fully exploited produced in al type of recycling process, authors mentioned

I consider that the section “2.1 About PET and Its Recycling Strategies” is a little bit longer than needed since the main objective of the review is the enzymes and their characteristics and applicability (Section 2.2)

Table 1. Reported PET-depolymerizing enzymes from marine microorganisms. Consider changing column information from “PET hydrolysis and reaction conditions” to “Yield” or “Hydrolysis performance” or so, to avoid mentioning all the physicochemical conditions in the column. Are the pH and NaCl concentrations really necessary in that column?

Both commentaries applied to sections 3.1 and 3.2. 

Author Response

We appreciate the reviewer’s constructive comments and suggestions on our manuscript. We have done our best to address all the feedback and revised the paper accordingly. The changes are highlighted in yellow, and detailed in our responses to each comment below.

Reviewer’s comment: Please consider the number of Keywords, 3-5 are generally ideal.

We reduced the number of Keywords to 5

Reviewer’s comment: Is there any possibility of including an abbreviations table? It would help the reader to consult along the document the use of abbreviations.

We have added an abbreviations’ list at the end of the manuscript, as suggested.

Reviewer’s comment: The introduction deals mainly with data regarding the production of plastics and a good panorama of the plastic life cycle.

In Lines 75-76 that “(i.e., monomers, oligomers, polymers) that can be used for various practical purposes, including creating value-added products” It would be interesting to address in this section more detail regarding the monomer’s characteristics or potential toxicity or the perspective that those monomers could be fully exploited produced in al type of recycling process, authors mentioned

Since plastic monomers vary significantly depending on the type of plastic (e.g., PET, PLA, PE), a detailed discussion of their characteristics, including potential toxicity and properties, would require an analysis that exceeds the scope of this review. Our focus is primarily on the enzymatic degradation and recycling potential of plastics, rather than an extensive exploration of the chemical properties of individual monomers. However, we agree that the potential for these monomers to be utilized across various recycling processes is an important consideration, and we have briefly addressed this in the text where relevant (lines 178-182).

Reviewer’s comment: I consider that the section “2.1 About PET and Its Recycling Strategies” is a little bit longer than needed since the main objective of the review is the enzymes and their characteristics and applicability (Section 2.2)

We believe the current length of Section 2.1, "About PET and Its Recycling Strategies," is essential for providing a thorough context for the review. This background enables readers to better understand the significance of enzymatic recycling technologies and their relevance to existing methods. As such, we prefer to retain the section as is to preserve the manuscript's flow and clarity.

Reviewer’s comment: Table 1. Reported PET-depolymerizing enzymes from marine microorganisms. Consider changing column information from “PET hydrolysis and reaction conditions” to “Yield” or “Hydrolysis performance” or so, to avoid mentioning all the physicochemical conditions in the column. Are the pH and NaCl concentrations really necessary in that column?

Both commentaries applied to sections 3.1 and 3.2. 

We consider that outlining the optimal conditions for enzyme catalysis is important for a thorough comparison of enzyme performance and enhances understanding of their potential applications. Therefore, we prefer to keep the current information in Table 1.
